# Tripodal Pd metallenes mediated by Nb$_2$C MXenes for boosting alkynes semihydrogenation

Zhongzhe Wei[1,3], Zijiang Zhao[1,3], Chenglong Qiu[1], Songtao Huang[1], Zihao Yao ®[1] ✉, Mingxuan Wang[1], Yi Chen[1], Yue Lin ®[2], Xing Zhong ®[1], Xiaonian Li[1] ✉ & Jianguo Wang ®[1] ✉

2D metallene nanomaterials have spurred considerable attention in heterogeneous catalysis by virtue of sufficient unsaturated metal atoms, high specific surface area and surface strain. Nevertheless, the strong metallic bonding in nanoparticles aggravates the difficulty in the controllable regulation of the geometry of metallenes. Here we propose an efficient galvanic replacement strategy to construct Pd metallenes loaded on Nb$_2$C MXenes at room temperature, which is triggered by strong metal-support interaction based on MD simulations. The Pd metallenes feature a chair structure of six-membered ring with the coordination number of Pd as low as 3. Coverage-dependent kinetic analysis based on first-principles calculations reveals that the tripodal Pd metallenes promote the diffusion of alkene and inhibit its overhydrogenation. As a consequence, Pd/Nb$_2$C delivers an outstanding turnover frequency of 10372 h$^{-1}$ and a high selectivity of 96% at 25 °C in the semihydrogenation of alkynes without compromising the stability. This strategy is general and scalable considering the plentiful members of the MXene family, which can set a foundation for the design of novel supported-metallene catalysts for demanding transformations.

Noble metal nanomaterials are one of the backbones of heterogeneous catalysis, possessing broad applications in the fine chemical and pharmaceutical industries[1–5]. To meet the demands of practical applications, extensive efforts are dedicated to engineering the morphology and electronic properties of diverse noble metal catalysts to minimize the metal loading and maximize the number of catalytic active sites[6–12]. Among them, 2D metallenes have spurred considerable investigations in heterogeneous catalytic reactions by virtue of sufficient unsaturated metal atoms, higher surface-to-volume ratio and surface strain[13–15]. The synthesis of metallenes mostly emphasizes the regulation of the thickness. Analogous to the structure of graphene, the controlled regulation of the geometry of metallenes with six-membered ring at the atomic level is largely unexplored. Besides, the metallene

nanomaterials are usually constructed by treating group VIII metal carbonyls with CO-releasing agents or organic ligands such as organic amines and surfactants[16]. An inevitable problem associated with the conventional metallenes is that in the final materials, difficult-to-remove templating agents inherited from the wet chemistry will block the active sites and veil the activity-structure relationship[14]. Meanwhile, the active metal atoms are easily oxidized or degraded, shortening the service life of the catalyst. Further, the current application scenarios of metallenes are limited, chiefly focusing on the electrocatalytic conversion of small molecules[17,18]. Therefore, it is imperative to develop an innovative metallene system to cope with these deficiencies and produce outstanding catalysts applied in thermo-catalysis to produce high-value-added fine chemicals on this basis.

[1]Institute of Industrial Catalysis, College of Chemical Engineering, Zhejiang University of Technology, Hangzhou 310032, PR China. [2]Hefei National Research Center for Physical Sciences at the Microscale, University of Science and Technology of China, Hefei 230026, China. [3]These authors contributed equally: Zhongzhe Wei, Zijiang Zhao. ✉e-mail: yaozihao@zjut.edu.cn; xnli@zjut.edu.cn; jgw@zjut.edu.cn

The pivotal challenge for controllable synthesis of metallenes is to break the strong metallic bonding in nanoparticles and lower the high surface energy of unsaturated coordination atoms in metallenes. Harnessing the strong metal-support interaction (SMSI)[19–21] could refrain from these puzzles, which is expected to become a promising strategy to form metallenes. The SMSI can effectively reduce the high surface energy of the metal nanoparticles for one thing[22–24]. Meanwhile, once the selected supports possess stronger interaction with metal atoms than the metal-metal binding, the supports would compel the metal atoms to diffuse on the support to form two-dimensional (2D) metal metallenes instead of three-dimensional (3D) metal nanoparticles, breaking the tradeoff between the catalytic stability and activity of traditional SMSI[25,26]. More importantly, given that the unique atom arrangement of 2D metal metallenes enabled faster mass transport and easier desorption of specific intermediate in the interlayer configuration[27], the application of 2D metal metallene can go far towards structure-sensitive reactions. The semihydrogenation of alkynes to alkenes is such a representative structure-sensitive reaction[24,28,29], which plays a crucial role in the industrial manufacture of polymers, pharmaceuticals, and vitamins and was chosen as a model reaction in this case study[30–33]. It has been recognized that the semihydrogenation of alkyne to alkene undergoes preferentially at the plane sites of Pd nanoparticles while the overhydrogenation of alkene occurs mainly at the edge sites due to the competitive adsorption of alkyne and alkene[29,34]. The ultrathin 2D metallene nanomaterials exposing specific facets such as (111) and (100) plane sites should be favor in the semihydrogenation of alkyne to the alkene. As a consequence, utilizing SMSI to provoke the in-situ growth of Pd metallenes on the support and further achieve the semihydrogenation of alkynes in a high activity, high selectivity, and high stability manner is a pivotal, yet challenging subject.

Herein we propose a facile galvanic replacement strategy to construct tripod Pd metallenes mediated by $Nb_2C$ MXenes at room temperature via SMSI accompanied by electron transfer from sub-surface Nb to Pd. Comprehensive characterizations, density functional theory (DFT) calculations and the molecular dynamic (MD) simulations evidence that the strong interaction between Pd and $Nb_2C$ with few functional groups triggers the formation of 2D Pd metallenes. Interestingly, the Pd metallenes exhibit a chair structure of six-membered ring likes cyclohexane, and the Pd atoms present a unique tripodal structure in which the upper Pd atoms are not bonded. Impressively, the 0.5 wt.% $Pd/Nb_2C$ catalyst with a low content of precious metal can yield a 96% selectivity and 10372 $h^{-1}$ TOF for the semihydrogenation of phenylacetylene, distinct from the Pd nanoparticles producing more byproduct of alkane. DFT calculations disclosed that the dilutive effect of surface Pd sites on Pd metallenes could significantly accelerate the diffusion of alkene, thereby boosting the semihydrogenation performance.

## Results

### MD simulations of Pd nanoparticles structure evolution over $Nb_2C$ and its derivatives

As the surface terminated groups could affect the properties of MXenes[13], ideal $Nb_2C$ with clean surface and $Nb_2C$ modified with functional groups (-Cl, -Br, and -O) were employed as models to explore the structural evolution of supported Pd nanoparticles by means of DFT calculations and MD simulations[26,35] (Fig. 1, Supplementary Figs. 1, 2). The force field parameters in the form of Morse potential were fitted based on the data of DFT calculations (Supplementary Table 1), and the fitting results accorded well with the DFT data. From snapshots of MD simulations, the supported $Pd_{561}$ nanoparticle has an obvious morphological tendency of 3D to 2D as the surface functional groups of -Cl, -Br, and -O on $Nb_2C$ turn into partial -O (Fig. 1). Further analysis found that the structure of nanoparticles

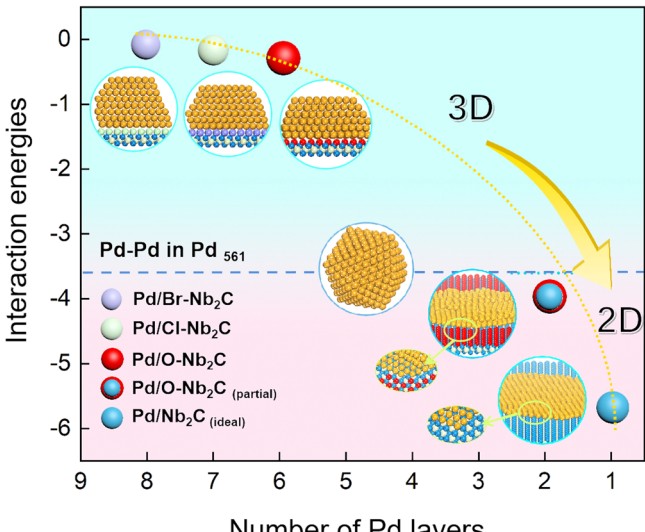

**Fig. 1 | The relationship of the interaction energies between Pd and supports and the number of Pd layers in the MD simulations of $Pd/MXenes$ catalysts.** Snapshots of $Pd_{561}$ nanoparticle supported on substrates of $Cl-Nb_2C$, $Br-Nb_2C$, $O-Nb_2C$, $O-Nb_2C_{(partial)}$ and $Nb_2C_{(ideal)}$ at 300 K. The palladium, chlorine, bromine, oxygen, carbon, and niobium atoms are represented as blue, green, brown red, red, gray, and dark green spheres, respectively.

strongly depends on the contest between Pd-Pd interaction and Pd-Nb interaction. On account of the strong Nb-Pd interaction for $Pd_{561}$ supported on ideal $Nb_2C$, the metal-support interaction manifestly surpasses the Pd-Pd interaction in nanoparticle, which makes the Pd nanoparticles exhibit a 2D planar structure (Supplementary Movies 1–3). Conversely, functional groups on the surface attenuate the metal-support interaction to less than the Pd-Pd interaction in nanoparticle, resulting in a 3D structure of Pd. Based on this, the targeted regulation of the structure of Pd nanoparticle can be achieved by manipulating the metal-support interaction. For $O-Nb_2C_{partial}$ (ideal $Nb_2C$ partially covered by O groups), at the beginning stage, a monolayer Pd was preferentially produced at the exposed Nb sites, which was then followed by the formation of a second layer of Pd once encountering oxygen functional groups, eventually exposing (111) facet. That is, $O-Nb_2C_{partial}$ can serve as a suitable support for the formation of Pd metallenes. This exciting result evoked us to exploit these catalysts experimentally, and further reveal the Pd structure-catalytic property relationship at the atomic scale.

### Structure characterization of a series of $Pd/Nb_2C$ catalysts

The $Nb_2C$ MXenes were synthesized by HF etching first and then dimethyl sulfoxide intercalation. The introducing of -Cl, -Br as well as -O groups on the surface of $Nb_2C$ ($Cl-Nb_2C$, $Br-Nb_2C$, $O-Nb_2C$) were easily accomplished by post-treatment of HCl, HBr and $H_2O_2$, respectively (Supplementary Figs. 3, 4). As the standard reduction potential of the $Nb^{5+}/Nb^{2+}$ is much lower than that of the $Pd^{2+}/Pd$ couple, the $Nb^{5+}/Nb^{2+}$ and $Pd^{2+}/Pd$ can be categorized as redox pairs[36,37]. The mixing of $PdCl_2$ with $Nb_2C$ MXenes triggers the spontaneous galvanic replacement reaction between $Pd^{2+}$ and reductive Nb species, thereby forming $Pd/Nb_2C$, $Pd/Cl-Nb_2C$, $Pd/Br-Nb_2C$, and $Pd/O-Nb_2C$. Pd content in the final catalysts is 0.49 wt.% determined by inductively coupled plasma-optical emission spectroscopy (ICP-OES), close to the target of 0.5 wt.%. The self-reduction process was monitored by XPS. After $Pd^{2+}$ was introduced on $Nb_2C$, the peak of Nb-C disappeared in Nb 3$d$ and C 1$s$ XPS spectra and the peak of Nb(V) oxide became better resolved in Nb 3$d$ and O 1$s$ spectra (Supplementary Fig. 5). The enrichment of $Pd^0$ and $Nb_2O_5$ on the surface of $Nb_2C$ confirmed the redox reaction between $Pd^{2+}$ and Nb species[38]. To probe the

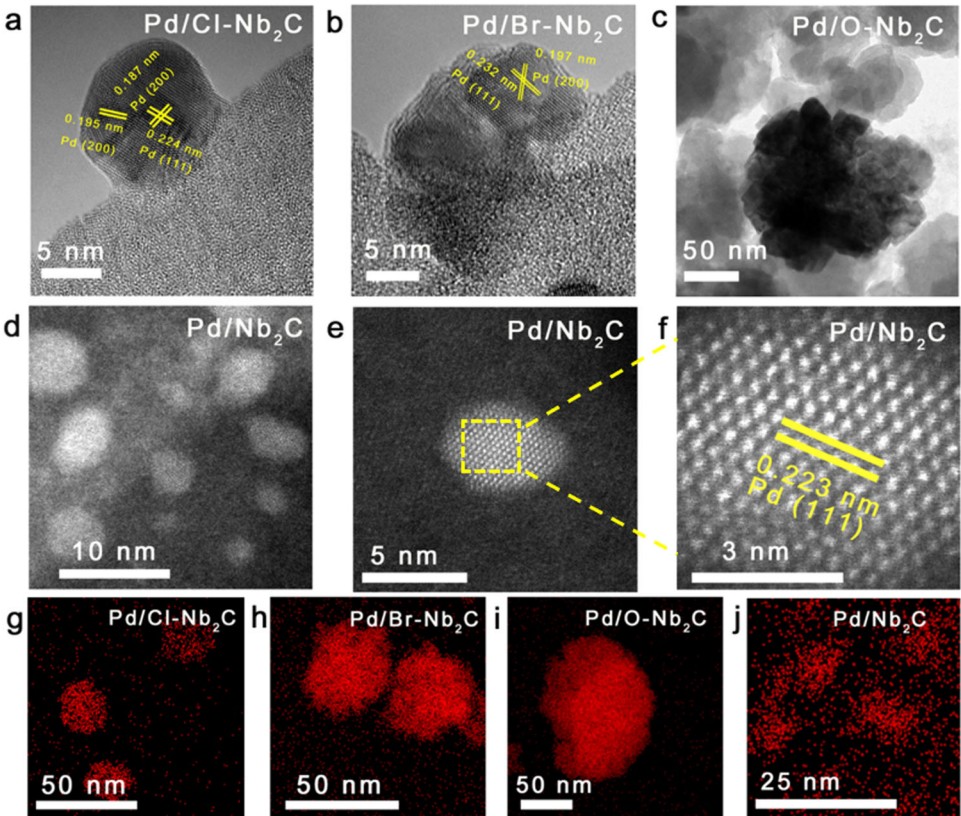

**Fig. 2 | Microstructure of Pd/Nb₂C MXenes. a–c** TEM images of Pd supported on functional groups-modified Nb₂C, **a** Pd/Cl-Nb₂C. **b** Pd/Br-Nb₂C. **c** Pd/O-Nb₂C. **d–f** HRTEM and AC-HAADF-STEM images of Pd/Nb₂C. **g–j** Elemental mapping of Pd/Cl-Nb₂C, Pd/Br-Nb₂C, Pd/O-Nb₂C and Pd/Nb₂C.

microstructural and electronic features of mentioned Pd/MXenes catalysts, a series of characterizations were undertaken. Scanning electron microscope (SEM) images showed that all the studied catalysts have a loose accordion-like structure besides Pd/O-Nb₂C (Supplementary Fig. 3). Pd presented different morphologies on the studied supports. The high-resolution transmission electron microscope (HRTEM) images manifested that well-faceted large Pd nanoparticles were anchored on Cl-Nb₂C, Br-Nb₂C and O-Nb₂C, displaying a rather wide range of sizes (roughly 10 nm to 1 μm) (Fig. 2a–c, Supplementary Figs. 6–8). The lattice spacing with 0.224 nm and 0.187 nm observed in Fig. 2a, b were attributed to the Pd (111) and Pd (200) facets. By sharp contrast, high-angle annular dark-field scanning transmission electron microscopy (HAADF-STEM) images (Fig. 2d; more images can be found in Supplementary Fig. 9) showed that the monodispersed Pd metallenes dispersed on Nb₂C presented an average lateral size of 6.2 nm. Furthermore, atomic-resolution HAADF-STEM revealed that the Pd metallenes showed lattice fringes with an interplanar spacing of 0.223 nm, which corroborated that the dominant exposed surfaces of Pd metallenes were (111) crystal facet (Fig. 2e, f). The energy dispersive X-ray spectroscopy (EDX)-mapping shown in Fig. 2g–j manifested that Pd species were dispersed as nanoparticles on functional groups-decorated Nb₂C, whereas as Pd metallenes on the Nb₂C. The atomic force microscopy (AFM) analysis was a valid and well-established method to determine the thickness of 2D materials[39,40]. As shown in Supplementary Fig. 10a, the uniform peaks with similar intensity spots in 3D tapping mode AFM image represented that Pd metallenes were homogeneously dispersed on the Nb₂C substrate. The corresponding 2D AFM images (Supplementary Fig. 10b) and height profiles (Supplementary Fig. 10c–e) showed that the thickness of the Pd metallenes was between 0.535 nm and 0.768 nm. Considering that the thickness of single-atom Pd (111) was 0.358 nm[40], it indicated that Pd metallenes were about two atomic

layers. Besides, the intensity of EDX line profiles was also related to the thickness of the materials[41]. The slightly undulating intensity of Pd exhibited in the EDX line profile (Supplementary Fig. 11) can be rationalized into the thin structure of Pd, which further implied the formation of Pd metallenes on Nb₂C. To further confirm the structure of the catalyst, the STEM image of a typical Pd metallene and the corresponding EDX elemental mappings of Pd (red), and O (blue) were presented in Supplementary Fig. 11. It can be clearly found that the Pd metallenes were surrounded by oxygen atoms, which was in agreement with MD simulations. Meanwhile, two DFT models with different locations of oxygen atom were designed and the calculated total energy indicated that oxygen atoms tended to be located at the interface of Pd and Nb₂C (Supplementary Fig. 12). To follow the growth processes of Pd on the Nb₂C, control experiments with different parameters, including self-reduction time and Pd loadings were further conducted. TEM images of 2%-Pd/Nb₂C and 5%-Pd/Nb₂C (Supplementary Fig. 13) showed that dendritic and petal-like Pd appeared, indicating the overgrowth of Pd. This observation verified that the decrease of Nb active sites would undermine the interaction with Pd, eventually leading to the agglomeration Pd on Nb₂C. To assess structural changes in the accordion-like Nb₂C after self-reduction of Pd, the X-ray diffraction (XRD) data of Nb₂AlC, Nb₂C and Pd/Nb₂C were collected (Supplementary Fig. 14). The appearance of (002) peaks in Nb₂C and the disappearance of the most intense nonbasal plane diffraction peaks in Nb₂AlC indicated that the MAX phases are converted to MXenes[42,43]. After the reduction of Pd ions, the (002) peak shifted to a higher angle, indicating that thin layers of Pd grew on the surface of Nb₂C. It is reasonable to speculate that the strong interaction between Nb₂C and Pd brings about the formation of Pd metallene structure.

Considering the noticeable morphological differences of Pd, it motivated us to implement X-ray absorption fine structure (XAFS) of Pd K-edge to reveal the local coordination environment of Pd in these

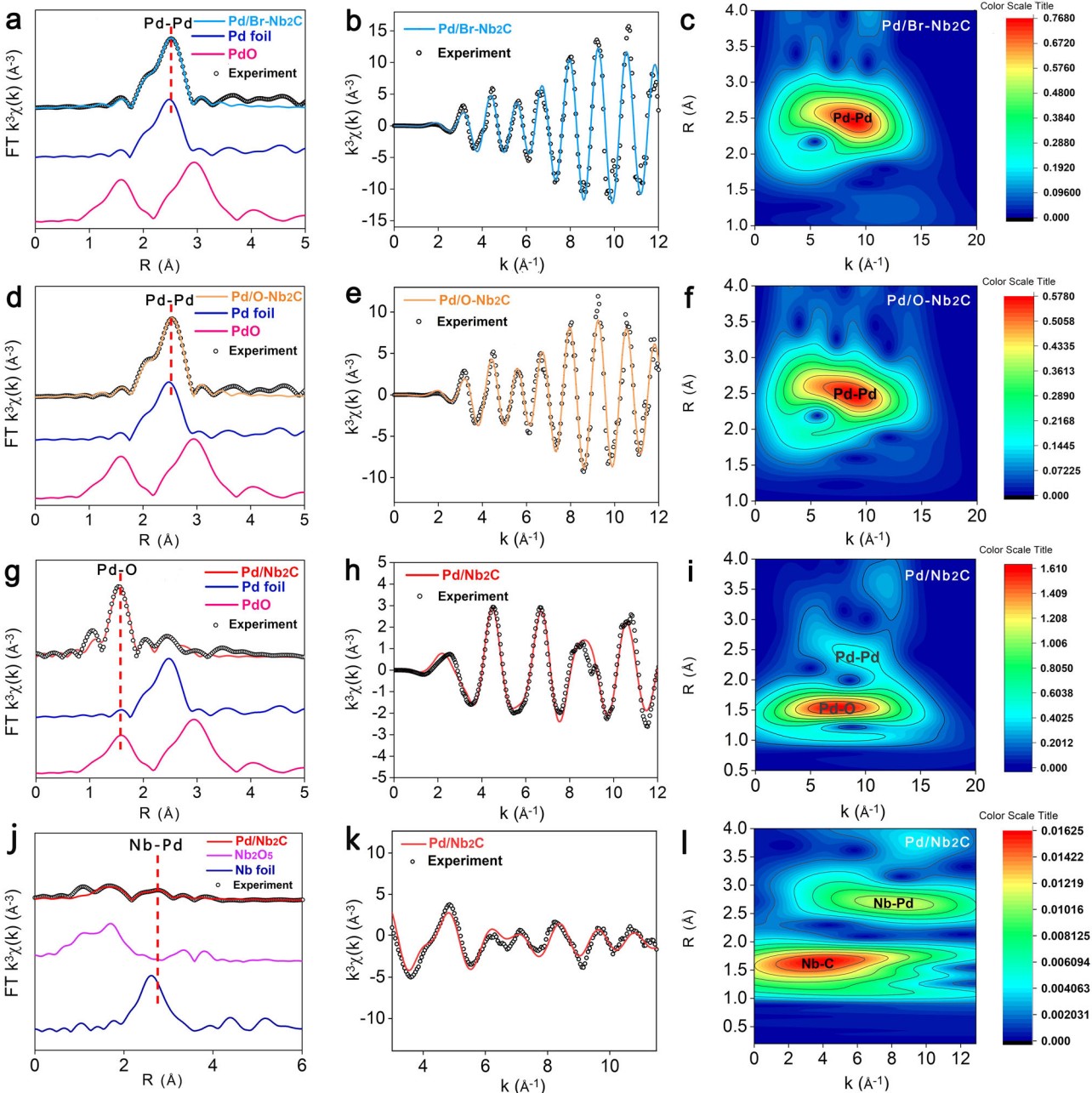

**Fig. 3 | Structures characterization of Pd/Nb₂C MXenes. a–c** XAFS analysis of Pd/Br-Nb₂C. **a** R-spaced FT-EXAFS of Pd *K*-edge. **b** k-space, and (**c**) WT-EXAFS plots of Pd. **d–f** XAFS analysis of Pd/O-Nb₂C, (**d**) R-spaced FT-EXAFS of Pd *K*-edge, (**e**) k-space, and (**f**) WT-EXAFS plots of Pd. **g–i** XAFS analysis of Pd/Nb₂C, (**g**) R-spaced FT-EXAFS of Pd *K*-edge, (**h**) k-space, and (**i**) WT-EXAFS plots of Pd. **j–l** XAFS analysis of Pd/Nb₂C, (**j**) R-spaced FT-EXAFS of Nb *K*-edge, (**k**) k-space, and (**l**) WT-EXAFS plots of Nb.

catalysts. In the Fourier transformed extended X-ray absorption fine structure (FT-EXAFS) data of Pd *K*-edge, an intensive coordination peak in R-space at about 2.75 Å and 2.76 Å is observed on Pd/Br-Nb₂C and Pd/O-Nb₂C, respectively, which could be attributed to the Pd-Pd scattering path (Fig. 3a, d, Supplementary Fig. 15). For Pd/Nb₂C, a relatively weak peak of Pd-Pd was located at about 2.74 Å (Fig. 3g). Besides the Pd-Pd peak, an additional prominent peak emerged at approximately 1.5 Å, which could be well fitted with the scattering interaction of Pd-O according to the PdO reference. The following average coordination numbers (CNs) of Pd according to EXAFS data-fitting were listed in Supplementary Table 2. CNs of Pd-Pd were fitted to be 11.1 and 10.4 in the Pd/Br-Nb₂C and Pd/O-Nb₂C, which were in good agreement with the results of particles overgrowth in HRTEM images. Notably, for Pd/Nb₂C, the first shell CN of Pd-O was about 2.1

and the second shell CN of Pd-Pd was 3. It is particularly interesting that, the average size of Pd species was measured to 6.2 nm, contradicting with its low CN and weak peak of Pd-Pd[44]. This unconventional phenomenon was highly relevant to the formation of an ultrathin Pd structure[45,46]. The EXAFS k-space spectra and corresponding model curves for three Pd/MXenes (Fig. 3b, e, h) showed similar oscillations in the low k (3−9 Å⁻¹) with R factor of 0.0179, 0.004 and 0.006, respectively. These values were all less than 0.02[47,48], indicating the accuracy of fitting results. The wavelet transform (WT) EXAFS (Fig. 3c, f, i) as a powerful technique was employed to distinguish these two bonds in the sample. The WT contour plots of Pd/Br-Nb₂C and Pd/O-Nb₂C showed a maximum lobe at 9.7 Å⁻¹ from 2.0 to 3.0 Å, which could index them to be Pd-Pd coordination. By contrast, a new lobe at about 7.61 Å⁻¹, 1.55 Å

corresponding to the Pd-O coordination in Pd/Nb$_2$C demonstrated that the surface Pd atoms adopted an unsaturated coordinated mode, again suggesting the successful fabrication of Pd metallenes loaded on Nb$_2$C.

To unveil the intrinsic mechanism of the formation of Pd metallenes, Nb $K$-edge XANES analysis was conducted and the results were presented in Fig. 3j–l and Supplementary Table 3. The peak at ~2.1 Å assigned to Nb-C confirmed that the successful preparation of Nb$_2$C[49,50]. It was worth noting that the peak of Nb-Pd was located at 2.72 Å (Fig. 3j). The fitting results revealed that the CN of Nb-Pd was 1.8, validating the existence for SMSI between Pd and Nb$_2$C. Since it is challenging to see the difference in the Nb $K$-edge EXAFS of Pd/Nb$_2$C, Pd/Br-Nb$_2$C and Pd/O-Nb$_2$C, WT analysis was also conducted to get more sight into this distinction (Supplementary Fig. 16). The Nb WT contour plot of Nb foil (Supplementary Fig. 15) afforded a forward lobe at (2.85 Å, 7.5 Å$^{-1}$), which could be ascribed to Nb−Nb coordination. Due to Pd has a higher atomic number, Nb-Pd back-scattering amplitude will move to higher $K$ value compared to Nb-Nb[51–53]. Consequently, the presence of lobe at (2.72 Å, 8.2 Å$^{-1}$) for Pd/Nb$_2$C shown in Fig. 3l can be ascribed to Nb−Pd coordination. Moreover, the Nb WT contour plots of Pd/O-Nb$_2$C and Pd/Br-Nb$_2$C recorded the shorter scattering range and a negative shift of the lobe, which was attributed to Nb-Nb bonds. Taken together, the above-mentioned results unambiguously ascertained that SMSI is the essence of the formation of Pd thin sublayers on Nb$_2$C, which were predicted by the MD simulations (Fig. 1).

Inspired by the EXAFS analysis and MD simulations, the atomic structure of tripodal Pd metallene was resolved by the state of art DFT calculations. Single-layer to four-layer Pd supported on Nb$_2$C were designed as models to investigate the possible structure. For the model of two-layer Pd loaded on Nb$_2$C, the simulated lattice space of Pd (0.222 nm) was close to the HRTEM observations with only a 0.4% error value (Supplementary Fig. 17). The Pd metallenes feature a chair structure of six-membered ring. The determined Pd atom in the top layer is only coordinated to three Pd atoms on the inside layer, nicely consistent with EXAFS results (Fig. 4a, b). The non-bond phenomenon was observed in the same top layer with a long Pd-Pd distance (0.302 nm) (Fig. 4a). This surface strain was induced by the lattice mismatch between Pd and the support Nb$_2$C.

As SMSI provides the prerequisite for the generation of Pd metallenes, XPS was then used to parse the metal-substrate interaction in Pd/Nb$_2$C by comparison with reference materials prepared with different reduction times[54]. The Pd 3$d$ XPS spectrum from Pd/Nb$_2$C is dominated by Pd$^0$ (89%), whose binding energies are located at 334.9 eV and 340.1 eV. During the progress of reduction, the Pd$^0$ peak distinctly shifts to lower binding energy (from 335.2 eV to 334.9 eV) whilst the Nb$_2$O$_5$ peak moves to higher binding energy[38] (Fig. 4d, e). Meanwhile, the proportion of Pd$^0$ increased steadily over time, from 61% to 89%, accompanied with an increased proportion of Nb$_2$O$_5$ from 43% to 80%. These results indicated the existence of charge transfer between Nb to Pd. CO stripping experiments also illustrated that the surface groups on the Nb$_2$C affect the capability of electronic transfer from Nb to Pd. The higher CO stripping potential of Pd/Nb$_2$C (0.91 V vs SCE) than Pd/Br-Nb$_2$C (0.79 V vs SCE) and Pd/Cl-Nb$_2$C (0.63 V vs SCE) indicated back-donation of electrons from Pd to CO 2$\pi^*$ antibonding orbitals was favorable for Pd/Nb$_2$C (Supplementary Fig. 18). This demonstrated that Pd in Pd/Nb$_2$C possess more populated 3$d$ state electrons[31,55], which is in accordance with the XPS results (Supplementary Fig. 19).

The electronic distribution between two-layer Pd (111) and Nb$_2$C was selected to further investigate the strong interaction properties by DFT calculations. It was found that the accumulation of charge was mainly localized in the upper Nb atoms and the inner layer Pd atoms. The negative charge was localized in the innermost Pd atoms, indicating the electrons were transferred from the upper Nb atoms (Fig. 4c). Specifically, the innermost Pd atom carried a negative charge of 0.24|e| averagely. In comparison with the inside Pd atom, the charge transfer from the inside Nb atom to the outer layer Pd atom was remarkable reduced to 0.05|e| averagely. In brief, the results of AC-HAADF-STEM, AFM, XAFS experiments and DFT calculations corroborate the successful production of Pd metallenes supported on Nb$_2$C via SMSI, as well as the evolution of Pd species on Nb$_2$C-based supports from 3D to 2D by altering the surface functional groups.

## Selective hydrogenation of phenylacetylene over Pd on Nb$_2$C and its derivatives

The different coordination environment of Pd will lead to significant differences in catalytic performances. The semihydrogenation of

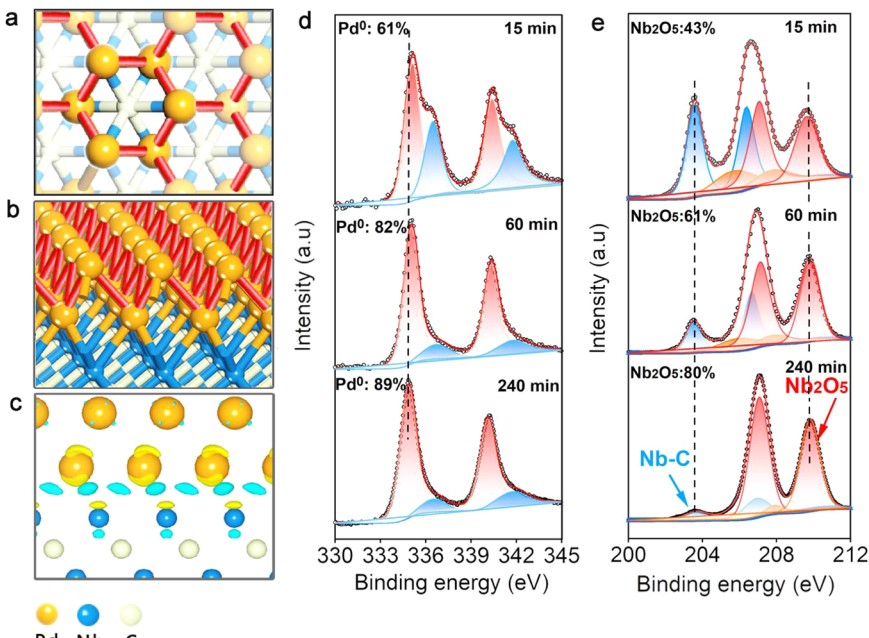

**Fig. 4 | Electronic properties of Pd/Nb$_2$C MXenes. a** The top view of simulated structure model of Pd/Nb$_2$C. **b** The oblique view of simulated structure model of Pd/Nb$_2$C. **c** Charge density difference of Pd atoms on Nb$_2$C. **d** XPS spectra of the Pd 3$d$ level of Pd/Nb$_2$C. **e** XPS spectra of the Nb 3$d$ level.

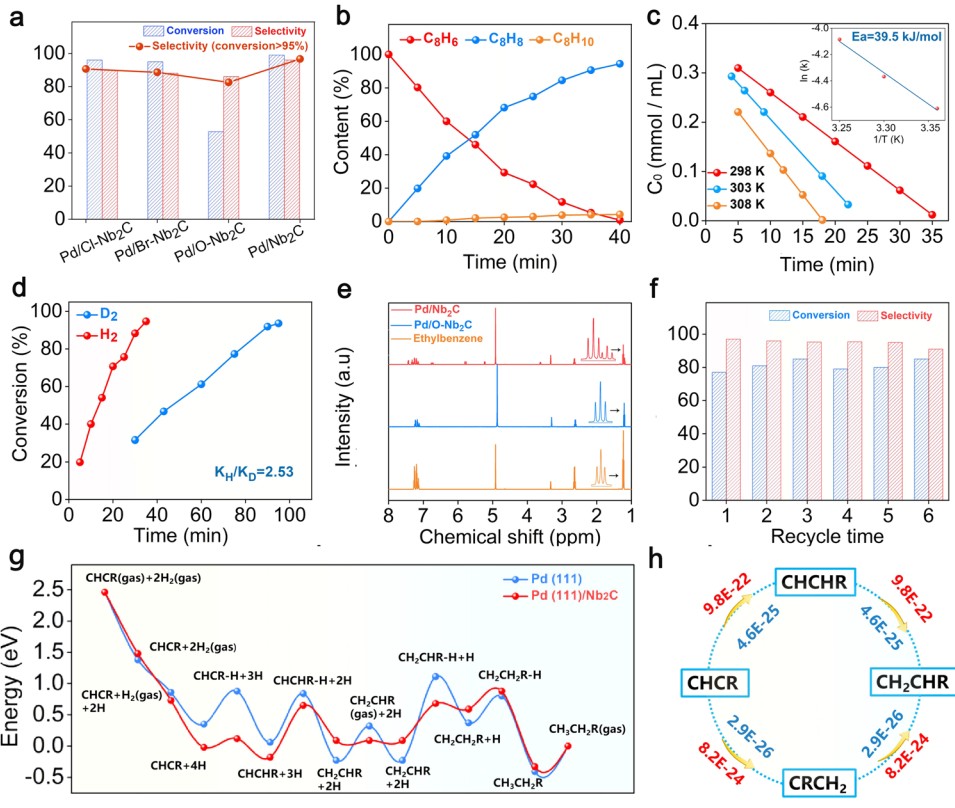

**Fig. 5 | Catalytic performance of Pd/Nb₂C MXenes. a** Catalytic performance of phenylacetylene on the different Pd/Nb₂C MXenes. Reaction conditions: 5 mL ethanol, 2 mmol phenylacetylene, 10 mg catalysts (0.023 mol % Pd), T = 298 K, H₂ pressure = 0.1 MPa. **b** Phenylacetylene hydrogenation reaction plots of Pd/Nb₂C. **c** Kinetic curves of Pd/Nb₂C at different temperatures and the activation energy. **d** Primary isotope effect observed for Pd/Nb₂C in phenylacetylene hydrogenation. **e** NMR data for the products of styrene hydrogenation carried out by different Pd catalysts in CD₃OD. **f** Catalytic stability for Pd/Nb₂C. **g** The DFT calculated coverage-dependent free energy diagram for the hydrogenation of phenylacetylene (CHCR) on Pd (111) and Pd/Nb₂C. **h** Simulated major pathways for the hydrogenation of CHCR. Reaction rates are also given for the associated molecular transformations. Reaction rates in red represent on Pd/Nb₂C. Reaction rates in blue mean on Pd (111). The values represent the reaction rates for each elementary step and are given in units of s⁻¹.

phenylacetylene was chosen as a probe reaction to evaluated the catalytic performance for a series of Pd/MXenes. Significantly, Pd/Nb₂C steered the reaction selectively towards styrene even at a conversion of 99% (96% selectivity) at 298 K with 0.1 MPa H₂ (Fig. 5a). In contrast to Pd/Nb₂C, the over-hydrogenation of styrene to ethylbenzene was more inclined to take place on Pd supported on Nb₂C modified with functional groups. Selectivities towards styrene on the Pd/Cl-Nb₂C, Pd/Br-Nb₂C and Pd/O-Nb₂C were 90%, 88%, and 82%, respectively. The product distribution with reaction time kinetic profile of Pd/Nb₂C revealed the good selectivity towards alkene throughout the reaction (Fig. 5b). Besides, the catalytic performance of phenylacetylene over Pd/Nb₂C with different loadings were also evaluated. With the increase of the Pd loading, the activity of the Pd/Nb₂C increased. The 5% Pd/Nb₂C can achieve 99% conversion in 7 min (Supplementary Fig. 20), but the corresponding selectivity has slightly decreased to 92%. The XRD pattern of 5% Pd/Nb₂C unfolded that no obvious characteristic peaks of Pd was detected (Supplementary Fig. 21). It was reasonable to speculate that the Pd petals were formed by stacking several layers of Pd metallenes rather than Pd nanoparticles. AFM images and height profiles confirmed the thickness of ~2 nm for petal Pd species (about five atomic layers) (Supplementary Fig. 22). Correspondingly, the morphology of more Pd atoms (Pd₉₂₃) on the Nb₂C surrounded by O group were further studied by MD simulations, which indicated that excess Pd atoms tended to accumulate on the top of Pd metallenes under the SMSI between Nb₂C and Pd (Supplementary Fig. 2).

Moreover, the TOF value for Pd/Nb₂C (10372 h⁻¹) is higher than Pd supported on other supports, such as activated carbon, metal oxides,

and it should be emphasized that it displayed an approximately 23-fold higher TOF value than traditional Lindlar catalyst (429 h⁻¹) (Supplementary Fig. 23). Reaction rates at different temperatures were measured and the activation energies (Eₐ) of Pd/Nb₂C were fitted to be 39.5 kJ/mol (Fig. 5c), suggesting the superiority of Pd/Nb₂C among Pd-based catalyst reported previously under similar reaction conditions[56,57]. To shed light on the original of excellent catalytic performance of Pd/Nb₂C, the experiments of kinetic isotope effect (KIE) were accomplished using D₂ as feed gas in phenylacetylene hydrogenation[58,59]. An obvious KIE was observed (ratio of reaction rates using H₂ and D₂, K_H/K_D = 2.53) (Fig. 5d), indicating heterolytic cleavage of H₂ might be involved in the hydrogenation process[60–62]. The nuclear magnetic resonance (NMR) experiments were conducted to verify this mechanism. The NMR data of the products on styrene hydrogenation was collected with different Pd catalysts in CD₃OD as the deuterium source and solvent (Fig. 5e). The orange peaks around 1.18 ppm corresponding to H atoms on α-C were split into two sets of triple red peaks, indicating that there was D transfer from the Pd/Nb₂C to β-C of ethylbenzene[58]. This phenomenon did not present on the Pd/O-Nb₂C (Fig. 5e, Supplementary Fig. 24a). Besides, the D substitution experiment of phenylacetylene was conducted using D₂O as deuterium source and isopropanol as solvent (Supplementary Fig. 24b). The peak around 3.47 belonging to H atoms on carbon-carbon triple bond of phenylacetylene decreased obviously after D substitution experiment. Above these results substantiated the heterolytic activation of H₂ yielded hydrogen proton at Pd/Nb₂C. It was generally accepted that heterolytic activation of H₂ would be related to the Pd-O interface.

According to the results of EXAFS for Pd/Nb$_2$C, the existence of Pd-O was the essential factor for heterolytic activation[58,63], again corroborating the formation of Pd metallene structure.

In terms of the stability of Pd/Nb$_2$C, we performed the selective hydrogenation of phenylacetylene at standard conditions in successive runs and compiled the results in Fig. 5f. Pleasingly, Pd/Nb$_2$C behaved efficiently during the reuse procedure, which can be recycled up to six consecutive runs with steady conversion (80%) and almost no selectivity deactivation (97%). After the durability test, the structure of the catalyst revealed no noticeable changes in size distribution (Supplementary Fig. 25). This phenomenon further highlights that Nb$_2$C can stabilize the Pd metallenes via SMSI.

As the accurate structure of Pd metallenes provides an ideal platform for the study of the structure-property relationship, DFT calculations were performed to further get insights into the molecular-level mechanisms of phenylacetylene hydrogenation over Pd/Nb$_2$C. The Pd (111) was selected as the benchmark catalyst compared with Pd/Nb$_2$C. First, the adsorption energies of phenylacetylene (CHCR), styrene (CH$_2$CHR) and ethylbenzene (CH$_3$CH$_2$R) on Pd (111) were much stronger than that on Pd/Nb$_2$C (Supplementary Fig. 26a). A closer study of the charge distribution of these molecules showed that the value of electrons was nearly the same on Pd (111) and Pd/Nb$_2$C, respectively, indicating electrons may not be the major factor affecting the adsorption energy of molecules. At the geometric configuration level, the longer dispersion distance (0.302 nm) between two adjacent Pd atoms led to the difference in the effective Pd atoms, substantially affecting the adsorption energy of intermediates. Herein, the effective Pd atoms represent the number of Pd atoms directly bonded with the molecules. From Pd (111) to Pd/Nb$_2$C, the effective Pd atoms in contact with CH$_2$CHR decreased from 6 to 5. The larger effective Pd atoms there are, the stronger the molecular adsorption. This phenomenon was also consistent with the weaker chemisorption energy of CHCR and CH$_3$CH$_2$R on Pd/Nb$_2$C compared to on pure Pd (111) (Supplementary Fig. 26b).

Because of the strong interaction between molecule and surface, the most challenging factor, coverage effects, should be taken into account. Based on our developed coverage-dependent model in previous work[64–66], the first detailed investigation of the reaction mechanisms for the hydrogenation of CHCR using the DFT-D3 functional and state-of-the-art microkinetic modeling[64,67,68] was explicitly carried out. All possible reaction channels on Pd (111) and Pd/Nb$_2$C (Supplementary Figs. 27–38) were investigated. The complete elementary steps are displayed in Supplementary Tables 4-8. With the coverage effect, the better diffusion ability of CH$_2$CHR (E$_{ad}$ ≈ 0 eV) on Pd/Nb$_2$C was observed compared to that on Pd (111) with the value of -0.55 eV. The high selectivity of CH$_2$CHR on Pd/Nb$_2$C can be explained by higher hydrogenation barrier of CH$_2$CHR + H → CH$_2$CH$_2$R (Ea = 0.58 eV) and CH$_2$CH$_2$R + H → CH$_3$CH$_2$R (Ea = 0.29 eV), which indicated that CH$_2$CHR is likely diffusion rather than further hydrogenation (Fig. 5g). Both models (coverage-dependent model and non-coverage model) indicated that the higher reaction rate of CH$_2$CHR formation was observed in the system of Pd/Nb$_2$C (Fig. 5h and Supplementary Fig. 39). The superiority of Pd/Nb$_2$C was also confirmed by simulating the hydrogenation of 2-methyl-3-butyn-2-ol (MBY) on Pd (111) and Pd/Nb$_2$C, respectively (Supplementary Figs. 40–44). These findings emphasize the critical role of tripodal Pd metallenes and explain why Pd/Nb$_2$C was far better than Pd (111) in boosting the catalytic performance, possibly profiting from the dilutive effect of Pd atoms in metallenes that could accelerate the diffusion of CH$_2$CHR.

### General Scope of Pd/Nb$_2$C
Having proved that Pd/Nb$_2$C was an efficient catalyst, subsequently, to explore the universality of catalysts, we tested the general scope of the catalyst for the hydrogenation of various structurally different substituted alkynes (Table 1, Supplementary Figs. 45–62 and

Supplementary Table 9). Gratifyingly, the Pd/Nb$_2$C was found to exhibit consistent selectivity for substituted alkynes. For terminal alkynes with alkyl, hydroxyl, ether, and amino substituents, Pd/Nb$_2$C delivered more than 90% selectivity towards terminal alkenes (Table 1, entries 1-11). Notably, terminal alkynes substituted with halogen groups (-F, -Cl, -Br) were readily hydrogenated to the desired alkenes and no dehalogenation products were detected (Table 1, entries 12-15). Pd/Nb$_2$C also exhibited high alkenes selectivity for the alkynes bearing biphenyl backbones, heterocyclic frameworks, and carbonyl functional groups (Table 1, entries 16-23). Impressively, reducible functional groups such as -NO$_2$, -CHO groups, remained completely unaffected during the hydrogenation process (Table 1, entries 24, 25). In addition to terminal alkynes, Pd/Nb$_2$C can accomplish the smooth

hydrogenation of internal alkynes to cis-alkenes with a selectivity of up to 96% for cis-2-buten-1-ol and 1-phenyl-1-propyne (Table 1, entries 26, 27). Similarly, Pd/Nb$_2$C displayed impressive chemoselectivity in the transformation of mifepristone to steroidal drug aglepristone (Table 1, entry 28), further indicating the potential application of Pd/Nb$_2$C in fine chemical industry. These results explicitly demonstrated that Pd/Nb$_2$C is a universal catalyst and displays superior selectivity in the hydrogenation of alkynes.

## Discussion
In summary, we have casted Pd metallenes supported on Nb$_2$C by exploiting spontaneous redox reaction between Nb$_2$C and Pd$^{2+}$ for boosting the semihydrogenation of alkynes. Through MD simulations and experimental verification, it was found that the transformation of 3D Pd nanoparticles to 2D Pd metallenes can be achieved by manoeuvring the metal-support interaction. The Pd metallenes feature a chair structure of six-membered ring with the coordination number of Pd as low as 3 due to the SMSI between Pd and Nb$_2$C. Pd/Nb$_2$C afforded an excellent TOF of 10372 h$^{-1}$ and a high selectivity to styrene of 96%. Advanced coverage-dependent kinetic analysis showed that the unique tripodal Pd metallenes enable the rapid desorption of alkenes, thus boosting the catalytic performance. The elaborately selected Nb$_2$C MXenes in this work not only acts as a template to guide the controllable growth of metallenes, but also stabilizes the Pd metallenes. This study will lead the design of novel supported-metallene catalysts in the advanced synthesis of refined chemical products.

## Methods
### Materials
Niobium aluminum carbide powder (Nb$_2$AlC, 90%, particle size 200 μm) was purchased from Forsman. Phenylacetylene (PA, 99.5%), palladium chloride (PdCl$_2$, >99%), hydrofluoric acid (HF, 40%), hydrochloric acid (HCl, 30%), hydrobromic acid (HBr, 48%), hydrogen peroxide (H$_2$O$_2$, 30%), dimethyl sulfoxide (DMSO, 99%) and other substituted alkyne compounds (99%), and Lindlar catalyst (5 wt % Pd) were purchased from Macklin. Unless otherwise stated, all solvents and chemicals were used without further treatment. All gases (N$_2$, H$_2$, D$_2$, CO) used for catalyst synthesis and catalytic reaction were ultra-high purity. Deionized water used in the experiment was obtained using a water purifier (HHitech).

### Preparation of Nb$_2$C
1 g Nb$_2$AlC powder was added into 30 mL (40%) of HF and stirred 48 h at 35 °C with 1000 rpm (111 × g). Subsequently, the suspension was separated from the mixture solution via centrifugation at 8500 rpm (8020 × g) for 5 min for removing HF. The obtained suspension was then re-dispersed into 25 mL DMSO and 20 mL H$_2$O, and stirred for 24 h at room temperature with 800 rpm (71.4 × g). Next, the mixture solution was separated again by centrifugation at 8500 rpm (8020 × g)

**Table 1 | The performance of Pd/Nb₂C catalyst toward different substrates**

| 1 | 2 | 3 | 4 | 5 |
|---|---|---|---|---|
| 94% (95%) | 96% (92%) | 95% (94%) | 96% (94%) | 96% (93%) |
| 6 | 7 | 8 | 9 | 10 |
| 92% (93%) | 99% (92%) | 97% (92%) | 96% (91%) | 92% (95%) |
| 11 | 12 | 13 | 14 | 15 |
| 99% (93%) | 99% (91%) | 94% (92%) | 98% (93%) | 96% (93%) |
| 16 | 17 | 18 | 19 | 20 |
| 92% (95%) | 93% (92%) | 99% (96%) | 88% (89%) | 99% (88%) |
| 21 | 22 | 23 | 24 | 25 |
| 87% (94%) | 99% (88%) | 99% (94%) | 99% (93%) | 99% (92%) |
| 26 | 27 | 28 | | |
| 96% (96%) | 97% (96%) | 97% (90%) | | |

**Reaction conditions:** alkyne substrates, catalyst of Pd/Nb₂C, 5 mL of ethanol. Detailed reaction conditions are listed in the Supplementary Information. Conversions were reported, and the data in parentheses were alkenes selectivity.

to remove DMSO. Finally, the Nb₂C was dispersed into 200 mL H₂O under the protection of N₂ and preserved in refrigerator at 5 °C.

### Synthesis of Pd/Nb₂C

Initially, 50 mL as-prepared Nb₂C solution (1.85 mg/mL) was added into 100 mL round bottom flask. Then, 0.77 mL of 1 mg/mL PdCl₂ was added to the Nb₂C solution under magnetic stirring for 4 h at 35 °C with N₂. Further, the Pd/Nb₂C precipitate was filtered and washed with distilled water. Finally, the obtained solid precipitate was dried in a vacuum oven at 60 °C overnight.

The synthetic procedures of Cl-Nb₂C, Br-Nb₂C, O-Nb₂C, Pd/Cl-Nb₂C, Pd/Br-Nb₂C, Pd/O-Nb₂C are presented in Supplementary Information. For Pd/Cl-Nb₂C and Pd/Br-Nb₂C, the contents of Cl and Br were

measured to be 2.52 wt. % and 1.1 wt. %, respectively according to the results of ICP-MS.

### Characterizations

The catalysts are characterized with by SEM (Hitachi, S-4800, Japan) with 15 kV accelerate voltage. TEM images were recorded with an electron microscope (Tecnai G2F30S-Twin) operated at 300 kV. HAADF-STEM and EDS mapping images were taken on a Thermo Fisher Titan Themis 60–300 "cubed" microscope, and the images were fitted with a series of aberration-correction factors for the imaging and probe forming lens.

XRD was performed on an X-ray diffractometer (PAN Alytical X-pert Pro) with Cu Kα irradiation (λ = 1.5418 Å) at 40 kV and 40 mA.

XPS measurements were performed in a Thermo Scientific K-Alpha. All BEs were referenced to the C 1$s$ peak at 284.6 eV of the surface adventitious carbon to correct the shift caused by charge effect. The actual loading of Pd were analyzed by ICP-OES using an Agilent 720ES.

XAS measurements were carried out at the XAS Beamline at the Australian Synchrotron (ANSTO) in Melbourne, Australia using a set of liquid nitrogen cooled Si (311) monochromator crystals. The electron beam energy is 3.0 GeV. With the associated beamline optics (Si-coated collimating mirror and Rh-coated focusing mirror), the harmonic content of the incident X-ray beam was negligible. A Ge 100 element detector was used to collect the fluorescence signal, and the energy was calibrated using a Pd foil and Nb foil. The beam size was about $1 \times 1$ mm. Note that a single XAS scan took ~1 h.

To probe the hydrogen transfer process using NMR, $^1$H NMR spectra were recorded in $CD_3OD$ on a Bruker Avance 600 MHz spectrometer.

### Electrochemical measurements

The CO stripping voltammetry of the catalysts were conducted on an electrochemical work station (Chenhua Instrument 760D, China) in a three-electrode cell at room temperature. 10 mg of catalyst, 15 μL of 5 wt% Nafion solution (Suzhou Yilongcheng Energy Technology Co., Ltd.) and 300 μL of dehydrated ethanol were mixed and sonicated for 30 min to acquired evenly mixture. The working electrodes were prepared by applying 10 μL of the homogeneous mixture on the surface of a carbon fiber electrode ($2 \times 2$ cm). A saturated calomel electrode (SCE) was used as reference electrode, and platinum wire served as counter electrode. CO gas was purged through the catalyst surface continuously in the cell filled with 0.5 M $H_2SO_4$ electrolyte for 30 min and 0.2 V versus SHE was imposed on the working electrode. Afterwards, the electrodes were quickly transferred to another cell filled with fresh $H_2SO_4$ electrolyte (without CO). The CO stripping curves were recorded in the potential range from 0.2 to 1.0 V (versus SCE) with a sweep rate of 2 mV s$^{-1}$.

### Catalytic tests

Selective hydrogenation of phenylacetylene was implemented in a 50 mL stainless-steel autoclave (Anhui Kemi Machinery Technology Co., Ltd). Typically, 2 mmol of phenylacetylene, 10 mg of 0.5% Pd/Nb$_2$C (0.023 mol % Pd), and 5 mL of ethanol were added in the autoclave in turn. Subsequently, the autoclave was purged with H$_2$ for three times, pressured to 0.1 MPa, and heated to 298 K. After the reaction, the remaining H$_2$ was discharged and the reaction mixture was obtained through centrifugation. The composition ratio of the mixture was analyzed by gas chromatography (GC), and the product composition was determined by gas chromatography-mass spectrometry (GC-MS) which was equipped with flame ionization detector (FID). The reaction time of recycle test was controlled at 20 min to ensure that the conversion reached about 80%. The catalyst was recovered by centrifugation, washed with ethanol for three times, and dried under vacuum at 333 K overnight and then used for the next run without additional reactivation.

H-D exchange experiments were carried out in a 50 mL stainless-steel autoclave at 298 K. 2 mmol of phenylacetylene, 10 mg of 0.5% Pd/Nb$_2$C (0.023 mol% Pd), and 5 mL of ethanol were added in the autoclave. Subsequently, the autoclave was purged with D$_2$ for three times, pressured to 0.1 MPa. The product composition was determined by GC-MS.

### Data availability

All data that support the findings of this study are included in the paper and the Supplementary Information, or are available from the corresponding author upon request.

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

## Acknowledgements

The authors acknowledge the financial support from the National Natural Science Foundation of China (22008213, U21A20298, 22141001), the Zhejiang Provincial Natural Science Foundation (LQ21B060005), the National Key Research and Development Project of China (2022YFE0113800), and the Fundamental Research Funds for the Provincial Universities of Zhejiang (RFA2022011).

## Author contributions

Z.Z.W., Z.J.Z., X.N.L., and J.G.W. conceived the project. Z.Z.W., Z.J.Z. contributed equally to this work. Z.Z.W. and Z.J.Z. designed catalysts and performed catalytic reactions. Z.H.Y. and C.L.Q. fulfilled the computational calculations. S.T.H., M.X.W., and Y.C. helped with the characterizations. Y.L. processed transmission electron microscopy analysis. Z.Z.W., Z.J.Z., Z.H.Y. and J.G.W. co-wrote the paper, X.N.L., and Z.Z. discussed the results. All authors commented on the manuscript.

## Competing interests

The authors declare no competing interests.
