## [Peer Review File · Nature Communications]

Tripodal Pd metallenes mediated by Nb₂C MXenes for boosting alkynes semihydrogenationREVIEWER COMMENTS

Reviewer #1 (Remarks to the Author):

This manuscript reports the synthesis of Pd metallenes on Nb₂C and its catalytic activity in the selective semi-hydrogenation of alkynes. The Pd metallene here is a two-layer, chair-like structure of ca. 6 nm planar diameter on the Nb₂C surface, where the original Nb(II) atoms have been oxidized to Nb₂O₅ while the Pd(II) atoms have been reduced to Pd(0). Kinetic and DFT calculations support that these “laminar” Pd structures facilitate the desorption of the hydrogenated alkyne intermediates to boost the reaction rate and selectivity during the semi-hydrogenation reaction.

The Pd-Nb₂C material is characterized by HR-TEM, XAS and XPS, among other techniques, and compared with homologous materials where the Nb₂C support has been previously treated with HCl, HBr and H₂O₂. The latter lead to the formation of well-defined Pd nanoparticles (NPs), which are catalytically (slightly) less selective, and less active. However, it must be said that the structure of the supported Pd metallene is not completely proved, experimentally, which perhaps explains why authors provide extensive MD calculations to support the structure. The HR-TEM photographs do not show any view of the zig-zag planar structure, and while it is true that the low coordination number of Pd in XAS support the planar structure, a TEM or AFM measurement confirming this point is missing. Another point which is not clear in the proposed structure is the location of the oxygen atoms. From the XPS measurements (and also XAS), it is clear that the Nb(II)-C bonds disappear and new Nb(V)-O are formed, thus, where are these oxygen atoms located? They are not displayed (or at least clearly indicated) even in the calculated structures.

Regarding the catalysis, this Reviewer thinks that the choice of the semi-hydrogenation of alkynes as the reaction of study, is good. I could not find any precedent with Pd metallenes and the authors' hypothesis on a better selectivity is adequate. However, some unclarified points must be solved. First, Fig S17 shows that the Lindlar catalyst has the poorer catalytic activity and selectivity among all catalysts tested, even the same or poorer selectivity than other supported Pd catalysts such as Pd-Al₂O₃ or Pd-TiO₂. These results are against all known literature. If this is so, the reaction conditions employed here are very special. Please revise this. What is the amount of Pd employed in these reactions? What is the excess of H₂? A detailed experimental procedure for all catalysts must be given.

Product characterization is not provided. While it is true that the obtained alkenes are well-known compounds, some data should be given at least for the internal alkynes (cis/trans). Here, the text claims that pure cis are obtained while the mechanism is heterolytic on the basis of H/D experiments, thus the possibility of forming trans alkynes is certain. Please check.

In summary, the use of Pd metallenes as catalysts for a chemo- and regio-selective organic reaction such as the semi-hydrogenation of alkynes seems a relevant new contribution to the catalysis field and suitable for publication in Nat. Commun. The catalytic use of Pd metallenes is an emerging field that, as far as I know, has been basically applied only in electrocatalytic transformations (authors cite a seminal work in Ref. 17 but new, very recent related works have been published). Thus, publication is recommended but only after the above-mentioned issues and those following below, are correctly addressed:

- The in-situ reduction of Pd salts on electronically rich carbon supports (even charcoal) is well-known,

for many years (i.e. J. Mol. Catal. A 230 (2005) 97–105). The use of the word “galvanic” here is perhaps not appropriate, it looks more an electrochemistry term, authors should explain it.

- Another weird concept to me is “ultra”-strong metal-support interaction (USMSI). I guess that the concept is brought here because the supported Pd NPs on, for instance, Cl-Nb₂C, clearly show SMSI (see HR-TEM photographs), which is by the way not clearly mentioned in the text. Please clarify the USMSI concept here.

- The amount of Br according to the XPS is much lower than Cl in the treated supports. Please explain. Any quantification?

- The text states that an increase in Pd loading produces further agglomeration, thus losses of Pd metallenes and catalytic activity. However, Fig S10 clearly shows that the yield and selectivity are exactly the same for the 0.3 wt% and the 5 wt% materials. The TEM photographs do not show, to me, higher agglomeration but, perhaps, Pd particle superpositions by the higher population of Pd metallenes. Could be that true? That will be good news, since lower amounts of solid will catalyze the reaction similarly. If not, and Pd nanoparticles are really formed, XRD measurements should show them.

- Color codes are missing in Figure 4 and Fig S14. Where are the oxygen atoms located?

- Figure 5: Figures e, g and I are too small to see anything. Please at least amplify them in the SI. Fig. S18: “i-proH” is not correct, please use i-PrOH or simply isopropanol. Substrate scope: “enol” is not a molecule but a functional group. It would be interesting here to test other hydrogenating functional groups besides Cl to test the selectivity of the Pd metallene catalyst, i.e. aldehyde, ketone or nitro group. Minor things: “Two layers” is “two layers” (several times across the SM), check some spacing in references.

Reviewer #2 (Remarks to the Author):

The submitted manuscript by deals with the use of experimental and theoretical approaches to deliver Pd metallenes on Nb₂C with a focus on boosting the semihydrogenation of alkynes (six-membered rings). On the brightest side, most of the experimental measures in the lab seem sound enough to exploit novel catalysts in the hydrogenation of alkyne. However, this referee feels that the computational strategy must be further refined is meaningful conclusions are sought. There are several major corrections to be addressed before recommending that work for publication in Nature Communications.

Computational comments.

1./ I guess that Figures 20 – 23 illustrates optimizations for phenylacetylene, am I right? Figures 24 – 28 are rather than confusing and do not help to compare structures vs. relative energy. However, a major issue appears in the computed profiles. The final release of CH₂CHR to the CH₂CHR(gas) counterpart is associated to a barrier of ca. 2 eV (ca. 190 kJ/mol). That barrier is not specifically discussed in Figure 29. In addition, all other computed barriers are in the range of 0.80 – 1.15 eV (77 – 111 kJ/mol). That numeric values significantly differ from the measured activation energy. As stated by the authors in p. 11 ‘Reaction rates at different temperatures were measured and the activation energies (E_a) of Pd/Nb₂C

were fitted to be 39.5 kJ/mol (Fig. 5c), suggesting the superiority of Pd/Nb₂C among Pd-based catalyst reported previously under similar reaction conditions.' Chemical model and level of theory must be cautiously revised to provide more accurate activation energies.

2./ Related to the previous comment, one would expect a more complete assessment of the alkyne semi-hydrogenation performance of the catalysts. In other words, thermodynamic corrections are missing in the present version.

3./ TOF values and the yields of products supported on the selected seven Pd catalysts demonstrated the impact of decoration. The quality and novelty of the paper might be increased by performing additional calculations, e.g., by including one additional substrate (i.e. Pd/V₂O₅, Pd/Al₂O₃) and/or by simulating one additional alkyne.

Experimental comments.

4./ Authors claimed that the reported results 'explicitly demonstrated that Pd/Nb₂C is a universal catalyst and displays superior selectivity in the hydrogenation of alkyne.' Unfortunately, the selected library of alkynes is too narrow to confirm such 'universality'. A wider panel of alkynes with a larger heterogeneity. This is a must to confirm that the methodology may also be applied to other types of alkynes, which is especially critical for problematic functionalities including ketones, aldehydes, heterocycles, nitrogen and sulfur-containing alkynes, to cite a few.

In short, the paper deals with a problem of interest and results are promising. However, additional work must be performed. Without such corrections, the paper is suitable for other journal, e.g., Journal of Catalysis or ChemCatChem.

Response to the referees' comments

Dear reviewers,

Thank you very much for taking your time to review our manuscript. Especially we want to express our appreciation for your advice and valuable proposals. The recommendations from you have been incorporated in the revision and we presented all of them in the following.

Referee 1

Comments:

This manuscript reports the synthesis of Pd metallenes on Nb₂C and its catalytic activity in the selective semi-hydrogenation of alkynes. The Pd metallenes here is a two-layer, chair-like structure of ca. 6 nm planar diameter on the Nb₂C surface, where the original Nb(II) atoms have been oxidized to Nb₂O₅ while the Pd(II) atoms have been reduced to Pd(0). Kinetic and DFT calculations support that these “laminar” Pd structures facilitate the desorption of the hydrogenated alkyne intermediates to boost the reaction rate and selectivity during the semi-hydrogenation reaction.

Comment 1:

The Pd-Nb₂C material is characterized by HR-TEM, XAS and XPS, among other techniques, and compared with homologous materials where the Nb₂C support has been previously treated with HCl, HBr and H₂O₂. The latter lead to the formation of well-defined Pd nanoparticles (NPs), which are catalytically (slightly) less selective, and less active. However, it must be said that the structure of the supported Pd metallene is not completely proved, experimentally, which perhaps explains why authors provide extensive MD calculations to support the structure. The HR-TEM photographs do not show any view of the zig-zag planar structure, and while it is true that the low coordination number of Pd in XAS support the planar structure, a TEM or AFM measurement confirming this point is missing.

Reply: Thank you very much for approving of our work and your careful guidance. According to the reported literature, the atomic force microscopy (AFM) analysis was a valid and well-established method to determine the thickness of 2D materials^{1,2}. Therefore, AFM was used to measure the thickness of Pd metallenes and the results were supplemented in **Supplementary Fig. 10**. The Pd metallenes on Nb₂C were identified by the 3D topographic atom imaging analysis. As shown in **Supplementary Fig. 10a**, the uniform peaks with similar intensity spots represented that Pd

metallenes were homogeneously dispersed on the Nb₂C substrate. The corresponding 2D AFM images and height profiles showed that the thickness of the Pd metallenes was between 0.535 nm to 0.768 nm. Considering that the thickness of single-atom Pd (111) was 0.358 nm³, it indicated that Pd metallenes were about two atomic layers. Besides, the intensity of HAADF-STEM and EDX line profiles were also related to the thickness of the materials. Therefore, EDX spectroscopy was employed in STEM mode to study the elemental distribution of Pd in Pd/Nb₂C. The EDX spectrum in **Supplementary Fig. 11e** showed the integrated pixel intensities of Pd metallenes corresponding to the EDX line scan section. The slightly undulating intensity of Pd exhibited in the EDX line profile can be rationalized into the thin structure of Pd, which further implied the formation of Pd metallenes on Nb₂C^{4,5}.

As for the structure of Pd in Pd/Nb₂C, AC-HAADF-STEM and AFM characterizations corroborated the formation of Pd metallenes on Nb₂C with the thickness of two atomic layers. In terms of the coordinated environment for Pd metallenes, XAFS experiments and DFT calculations displayed that the coordination number of Pd was 3, where the upper Pd atom coordinates with the lower three Pd atoms. Therefore, the coordination structure of Pd can be described as zig-zag structure graphically. However, the coordination structure of zig-zag was difficult to clearly discern for the following reasons: (1) The stacking of layered Pd atoms would obscure the zig-zag coordination structure to a large extent. (2) Due to the atomic number and relative atomic mass of Pd are close to that of Nb in the system of Pd/Nb₂C, it is challenging to distinguish the zig-zag coordination structure from HAADF-STEM using Z-contrast and EDX mapping. In the manuscript, the two atomic layers of Pd metallenes was comprehensively verified by AC-HAADF-STEM and AFM characterizations, and the coordination structure of Pd metallenes were unambiguously confirmed by XAFS and DFT calculations.

As the comments by the reviewer, the manuscript is revised as follow:

◆ In the manuscript

Furthermore, atomic-resolution HAADF-STEM revealed that the Pd metallenes showed lattice fringes with an interplanar spacing of 0.223 nm, which corroborated that the dominant exposed surfaces of Pd metallenes were (111) crystal facet (**Fig. 2f**). The energy dispersive X-ray spectroscopy (EDX)-mapping shown in **Figs. 2g-i** manifested that Pd species were dispersed as nanoparticles on functional groups-decorated Nb₂C, whereas as Pd metallenes on the Nb₂C. **The atomic force**

microscopy (AFM) analysis was a valid and well-established method to determine the thickness of 2D materials^{39,40}. As shown in **Supplementary Fig. 10a**, the uniform peaks with similar intensity spots in 3D tapping mode AFM image represented that Pd metallenes were homogeneously dispersed on the Nb₂C substrate. The corresponding 2D AFM images (**Supplementary Fig. 10b**) and height profiles (**Supplementary Fig. 10c-e**) showed that the thickness of the Pd metallenes was between 0.535 nm to 0.768 nm. Considering that the thickness of single-atom Pd (111) was 0.358 nm⁴⁰, it indicated that Pd metallenes were about two atomic layers. Besides, the intensity of EDX line profiles was also related to the thickness of the materials⁴¹. The slightly undulating intensity of Pd exhibited in the EDX line profile (**Supplementary Fig. 11e**) can be rationalized into the thin structure of Pd, which further implied the formation of Pd metallenes on Nb₂C.

Added Figures in Supplementary Information

Supplementary Fig. 10 | AFM images of Pd/Nb₂C. (a) 3D tapping mode AFM image of Pd

metallenes. (b) 2D AFM image of Pd/Nb₂C corresponding to the tapping mode AFM image. (c, d, e) Height profiles along the three marked lines in (b).

Supplementary Fig. 11 | Elemental mappings of Pd/Nb₂C and EDX scan line. (a) HAADF-STEM image of Pd/Nb₂C. Elemental mapping of (b) Pd, (c) O and (d) composite mapping of Pd vs O. (e) Integrated pixel intensities of Pd (red) and Nb (blue) in Pd/Nb₂C.

Reference

1. Duan, H., *et al.* Ultrathin rhodium nanosheets. *Nat. Commun* **5**, 3093 (2014). <https://doi.org/10.1038/ncomms4093>
2. Li, X., *et al.* PdFe Single-Atom Alloy Metallene for N₂ Electroreduction. *Angew. Chem. Int. Ed* **61**, e202205923 (2022). <https://doi.org/10.1002/anie.202205923>
3. Jiang, J., Ding, W., Li, W., Wei, Z. Freestanding Single-Atom-Layer Pd-Based Catalysts: Oriented Splitting of Energy Bands for Unique Stability and Activity. *Chem* **6**, 431-447 (2020). <https://doi.org/10.1016/j.chempr.2019.11.003>
4. Jia, Z., *et al.* Fully-exposed Pt-Fe cluster for efficient preferential oxidation of CO towards hydrogen purification. *Nat. Commun.* **13**, 6798 (2022). <https://doi.org/10.1038/s41467-022-34674-y>
5. Xiong, Y., Yang, Y., Disalvo, F. J., Abruña, H. D. Pt-Decorated Composition-Tunable Pd-Fe@Pd/C Core-Shell Nanoparticles with Enhanced Electrocatalytic Activity toward the Oxygen Reduction Reaction. *J. Am. Chem. Soc.* **140**, 7248-7255 (2018). <https://doi.org/10.1021/jacs.8b03365>

Comment 2:

Another point which is not clear in the proposed structure is the location of the oxygen atoms. From the XPS measurements (and also XAS), it is clear that the Nb(II)-C bonds disappear and new Nb(V)-O are formed, thus, where are these oxygen atoms located? They are not displayed (or at least clearly

indicated) even in the calculated structures.

Reply: Thank you for your valuable comments. The location of oxygen atoms in Pd/Nb₂C had been confirmed by EDX spectroscopy and theoretical calculations.

To demonstrate the structure of Pd/Nb₂C, HAADF-energy dispersive X-ray (EDX) spectroscopy was employed to study the elemental distribution of Pd and O in Pd/Nb₂C and the results were added in Supplementary Information. **Supplementary Fig. 11** presented the STEM image of a typical Pd MXene and the corresponding EDX elemental mappings of Pd (red), and O (blue). It can be clearly found that there was an obvious boundary between O and Pd, and the oxygen atoms were located at the interface between Pd metallene and Nb₂C, which was in agreement with MD simulation (**Fig. R1**).

To further confirm the location of oxygen atoms in the catalyst, two DFT models with different locations of oxygen atoms were designed. Model 1 was the distribution of oxygen atom on the surface of Pd, and Model 2 was the distribution of oxygen atom at the interface of Pd and Nb₂C. The total energy was calculated to test the stability of the two DFT models (**Supplementary Fig. 12**). The results showed that the total energy of Model 2 was 3.6 eV lower than that of Model 1, indicating that the structure of Model 2 was more stable than Model 1 and oxygen atoms tended to be located at the interface of Pd and Nb₂C.

Fig. R1. MD simulation of Pd metallenes on the O-Nb₂C_{partial}.

As the comments by the reviewer, the manuscript is revised as follow:

◆ In the manuscript:

To further confirm the structure of the catalyst, the STEM image of a typical Pd MXene and the corresponding EDX elemental mappings of Pd (red), and O (blue) were presented in **Supplementary**

Fig. 11. It can be clearly found that the Pd metallene were surrounded by oxygen atoms, which was in agreement with MD simulations (**Fig. 1**). Meanwhile, two DFT models with different locations of oxygen atom were designed and the calculated total energy indicated that oxygen atoms tended to be located at the interface of Pd and Nb₂C (**Supplementary Fig. 12**). To follow the growth processes of Pd on the Nb₂C, control experiments with different parameters, including self-reduction time and Pd loadings were further conducted. TEM images of 2%-Pd/Nb₂C and 5%-Pd/Nb₂C (**Supplementary Fig. 13**) showed that dendritic and petal-like Pd appeared, indicating the overgrowth of Pd.

Added Figures in Supplementary Information

Supplementary Fig. 11 | Elemental mappings of Pd/Nb₂C and EDX scan line. (a) HAADF-STEM image of Pd/Nb₂C. Elemental mapping of (b) Pd, (c) O and (d) composite mapping of Pd vs O. (e) Integrated pixel intensities of Pd (red) and Nb (blue) in Pd/Nb₂C.

Supplementary Fig. 12 | Two DFT models of Pd/Nb₂C. (a) O atom on the surface of Pd in Pd/Nb₂C. (b) O atom at the interface of Pd and Nb₂C.

Comment 3:

However, some unclarified points must be solved. First, Fig S17 shows that the Lindlar catalyst has the poorer catalytic activity and selectivity among all catalysts tested, even the same or poorer selectivity than other supported Pd catalysts such as Pd-Al₂O₃ or Pd-TiO₂. These results are against all known literature. If this is so, the reaction conditions employed here are very special. Please revise this. What is the amount of Pd employed in these reactions? What is the excess of H₂? A detailed experimental procedure for all catalysts must be given.

Reply: Thank you for your careful comment. Through a lot of literature research, we found that different types of Lindlar catalysts may exhibit different catalytic activities. Liang and co-authors reported that the Lindlar catalyst showed very low catalytic activity (2.71% conversion of phenylacetylene) over 20 min under mild reaction conditions (303 K, 0.6 MPa)⁶. In some cases, the catalytic activity of Lindlar catalyst was poorer than that of Pd/Al₂O₃ and Pd/TiO₂. It had been reported that Pd/Al₂O₃-based catalysts exhibited higher catalytic performance than Lindlar catalyst⁷. In terms of Pd/TiO₂, it showed higher selectivity than commercial Lindlar catalyst, which was benefited from the SMSI between Pd and TiO₂⁸. The Lindlar catalyst used in this work was 5% Pd loaded on CaCO₃ with Pb poisoning, which was purchased from Aladdin. 0.5% Pd/Al₂O₃ and 0.5% Pd/TiO₂ were synthesized by NaBH₄ reduction method. The detailed preparation process of the control catalysts was supplemented in the Supplementary Information.

In **Supplementary Fig. 23**, the Pd employed in selective hydrogenation of phenylacetylene was 0.00047 mmol and the molar ratio of Pd to phenylacetylene is 0.023 mol%. In particular, for the Lindlar catalyst, the molar ratio of Pd to phenylacetylene is 0.23 mol%. In all hydrogenation tests, the reaction temperature was 298 K and the H₂ pressure was 0.1 MPa. It was reasonable to deduce that the low selectivity and activity of Lindlar catalyst was attributed to the rather mild reaction conditions. To illustrate this, the catalytic hydrogenation of phenylacetylene using Lindlar catalyst was conducted under harsh conditions with quinoline additive. It was found that it exhibited higher selectivity (93%) than 0.5% Pd/Al₂O₃ and 0.5% Pd/TiO₂ (**Table R1**).

Table R1. Catalytic hydrogenation performance of Lindlar catalyst under different reaction conditions ^a

Entry	Catalyst	Pressure (MPa)	Temperature (K)	Time (min)	Conv. (%)	Sel. (%)
1	Lindlar	0.5	333	25	94	88
2 ^b	Lindlar	0.5	333	40	91	93
3	Lindlar	0.1	298	40	44	91

^a Reaction conditions: 2 mmol of phenylacetylene, 10 mg of Lindlar catalyst (0.23 mol % Pd), 5 mL of ethanol, 1000 rpm. ^b 5 μ L quinoline additive.

As the comments by the reviewer, the manuscript is revised as follow:

◆ In the Supplementary Information

Added Text in Supplementary Methods

Synthesis of Pd/Al₂O₃. Initially, 200 mg γ -Al₂O₃ was dispersed into 50 mL deionized water in the bottom flask. Then, 1.7 mL of 1 mg/mL PdCl₂ aqueous solution was added to the mixture of γ -Al₂O₃ solution under magnetic stirring for 10 min at 25 °C. Further, 10 mL NaBH₄ aqueous solution (20 mg/mL) was added into above solution dropwise. Next, the 0.5% Pd/Al₂O₃ precipitate was filtered and washed with double distilled water. Finally, the obtained solid precipitate was dried in a vacuum oven at 60 °C overnight.

Synthesis of Pd/TiO₂. Initially, 200 mg P25-TiO₂ was dispersed into 50 mL deionized water in the bottom flask. Then, 1.7 mL of 1 mg/mL PdCl₂ aqueous solution was added to the mixture of P25-TiO₂ solution under magnetic stirring for 10 min at 25 °C. Further, 10 ml NaBH₄ aqueous solution (20 mg/mL) was added into above solution dropwise. Next, the 0.5% Pd/TiO₂ precipitate was filtered and washed with double distilled water. Finally, the obtained solid precipitate was dried in a vacuum oven at 60 °C overnight.

Revised figure in Supplementary Information:

Supplementary Fig. 23 | TOF values and the yields of products for various supported Pd catalysts. Conditions: 2 mmol of phenylacetylene, 10 mg of catalyst with 0.5 wt. % Pd (0.023 mol % Pd), 5 mL of ethanol, 298 K, 0.1 MPa of H₂, 40 min. ^a The dosage of Lindlar catalyst (5% wt. % Pd loaded on CaCO₃ with Pb poisoning) was 10 mg (0.23 mol % Pd).

Reference

- Wang, Z.-S., Yang, C.-L., Xu, S.-L., Nan, H., Shen, S.-C., Liang, H.-W. Electronic Modulation of Pd-Based Bimetallic Catalysts with Sulfur-Doped Carbon Support for Phenylacetylene Semihydrogenation. *Inorg. Chem.* **59**, 5694-5701 (2020). <https://doi.org/10.1021/acs.inorgchem.0c00458>
- Cordoba, M., Coloma-Pascual, F., Quiroga, M. E., Lederhos, C. R. Olefin Purification and Selective Hydrogenation of Alkynes with Low Loaded Pd Nanoparticle Catalysts. *Ind. Eng. Chem. Res.* **58**, 17182-17194 (2019). <https://doi.org/10.1021/acs.iecr.9b02081>
- Weerachawanask, P., Mekasuwandumrong, O., Arai, M., Fujita, S.-I., Praserttham, P., Panpranot, J. Effect of strong metal-support interaction on the catalytic performance of Pd/TiO₂ in the liquid-phase semihydrogenation of phenylacetylene. *J. Catal.* **262**, 199-205 (2009). <https://doi.org/10.1016/j.jcat.2008.12.011>

Comment 4:

Product characterization is not provided. While it is true that the obtained alkenes are well-known compounds, some data should be given at least for the internal alkynes (cis/trans). Here, the text claims that pure cis are obtained while the mechanism is heterolytic on the basis of H/D experiments, thus the possibility of forming trans alkynes is certain. Please check.

Reply: Thanks a lot for your critical question. We are sorry that the description of selectivity of cis-alkenes in original Table 1 was controversial and led to your misunderstanding. Just as the reviewer

proposed, for the selective hydrogenation of internal alkynes on Pd/Nb₂C, trans-alkenes were obtained as byproduct. As shown in **Table 1**, the selectivity of Pd/Nb₂C toward cis-2-butene-1-ol and trans-2-butene-1-ol was 96% and 2.3%, respectively in the hydrogenation of substrate 26. The selectivity toward cis-1-phenyl-1-propene and trans-1-phenyl-1-propene was 96% and 2.5%, respectively for the hydrogenation of substrate 27. According to the reported literature, for the stereoselectivity hydrogenation of alkynes, cis-alkenes and trans-alkenes were usually detected by GC-MS⁹. Therefore, this work adopted GC-MS characterization to determine the cis-alkenes and trans-alkenes. The detailed heating procedure for chromatographic column in GC-MS was as follows: elevating the temperature from 50 °C to 200 °C for 2 min at a ramping rate of 5 °C min⁻¹. To further confirm the product, the standard cis-2-butene-1-ol (purity: 98%) and standard cis-1-phenyl-1-propene (purity: 98%) were purchased and then detected with GC-MS, which were compared with the products in the hydrogenation of substrate 26, 27 (**Fig. R2, R3**). The standard cis-2-butene-1-ol and cis-1-phenyl-1-propene were purchased from the Suzhou Biolight company and Shanghai Aladdin company, respectively. The gas chromatographic peak and mass spectrum of the standard samples were consistent and the products, which confirmed that the main products were cis-alkenes. These results further highlighted the high selectivity of Pd/Nb₂C for the hydrogenation of internal alkynes.

Fig. R2. GC-MS of standard-cis-2-buten-1-ol and products for hydrogenation of substrate 26 at high conversion.

Fig. R3. GC-MS of standard-cis-1-phenyl-1-propene and products for hydrogenation of substrate 27 at high conversion.

Reference

9. Tokmic, K., Fout, A. R. Alkyne Semihydrogenation with a Well-Defined Nonclassical Co-H₂ Catalyst: A H₂ Spin on Isomerization and E-Selectivity. *J. Am. Chem. Soc.* **138**, 13700-13705 (2016). <https://doi.org/10.1021/jacs.6b08128>

Comment 5:

The in-situ reduction of Pd salts on electronically rich carbon supports (even charcoal) is well-known, for many years (i.e. *J. Mol. Catal. A* 230 (2005) 97-105). The use of the word “galvanic” here is perhaps not appropriate, it looks more an electrochemistry term, authors should explain it.

Reply: Thanks a lot for your enlightening comment. The word “galvanic” has been frequently used in reported literature¹⁰⁻¹². The article you mentioned proposes a novel spontaneous reduction of Pd²⁺ to Pd(0) on the single wall carbon nanotubes (SWNT)¹³ and we had cited this interesting work. Similarly, many catalysts, such as Pt/SWNT, Au/SWNT, Pd/GO, Pd/GDYO, Pt/DG, Ir/TNT and so on were all synthesized via redox reaction without the aid of a reducing agent. The mechanism is to take

advantage of the high work function of support in giving an energy of the Fermi level high enough to reduce the M^{n+} . Moreover, the electrochemistry term of ‘galvanic’ was also mentioned in the synthesis of CuAu/MWNT, NiAu alloys, RuNi single atom alloy, Pd/Cu(111) and Pd@Ru nanosheets. Actually, the metals with higher reduction potential (SHE, standard hydrogen electrode) can reduce metal ions with lower reduction potential, which was considered to be a galvanic displacement reaction.

In this work, given that Nb^{5+}/Nb^{2+} has a redox potential of -0.46 V vs SHE (standard hydrogen electrode), lower than that of Pd^{2+} (+0.62 V vs SHE), which suggested that Nb_2C can reduce Pd^{2+} into Pd^0 via a galvanic displacement reaction. That is, the Nb^{5+}/Nb^{2+} and Pd^{2+}/Pd can be categorized as such redox pairs. The relative potential levels rationalized the spontaneous electron transfer from the Nb^{2+} (oxidation) to the Pd^{2+} ions and their reduction. The mechanism of Pd deposition on Nb_2C is essentially the same as that on SWNT, both of which belong to the application of galvanic displacement strategy. Therefore, it was reasonable to use “galvanic” to describe the reduction process.

As the comments by the reviewer, the manuscript is revised as follow:

◆ In the manuscript:

As the standard reduction potential of the Nb^{5+}/Nb^{2+} is much lower than that of the Pd^{2+}/Pd couple, the Nb^{5+}/Nb^{2+} and Pd^{2+}/Pd can be categorized as redox pairs^{36,37}. The mixing of $PdCl_2$ with Nb_2C MXenes triggers the spontaneous galvanic replacement reaction between Pd^{2+} and reductive Nb species, thereby forming Pd/Nb_2C , $Pd/Cl-Nb_2C$, $Pd/Br-Nb_2C$ and $Pd/O-Nb_2C$.

Reference

10. Choi, H. C., Shim, M., Bangsaruntip, S., Dai, H. Spontaneous Reduction of Metal Ions on the Sidewalls of Carbon Nanotubes. *J. Am. Chem. Soc.* **124**, 9058-9059 (2002). <https://doi.org/10.1021/ja026824t>
11. Bruno, J. E., *et al.* Supported Ni–Au Colloid Precursors for Active, Selective, and Stable Alkyne Partial Hydrogenation Catalysts. *ACS Catal.* **10**, 2565-2580 (2020). <https://doi.org/10.1021/acscatal.9b05402>
12. Qu, L., Dai, L. Substrate-Enhanced Electroless Deposition of Metal Nanoparticles on Carbon Nanotubes. *J. Am. Chem. Soc.* **127**, 10806-10807 (2005). <https://doi.org/10.1021/ja053479+>
13. Corma, A., Garcia, H., Leyva, A. Catalytic activity of palladium supported on single wall carbon nanotubes compared to palladium supported on activated carbon: Study of the Heck and Suzuki couplings, aerobic alcohol oxidation and selective hydrogenation. *J. Mol. Catal. A.* **230**, 97-105 (2005). <https://doi.org/10.1016/j.molcata.2004.11.030>

Comment 6:

Another weird concept to me is “ultra”-strong metal-support interaction (USMSI). I guess that the

concept is brought here because the supported Pd NPs on, for instance, Cl-Nb₂C, clearly show SMSI (see HR-TEM photographs), which is by the way not clearly mentioned in the text. Please clarify the USMSI concept here.

Reply: Thanks a lot for your inspiring comment. This work proposes the concept of ultra-strong metal-support interaction (USMSI) to distinguish it from traditional SMSI.

Due to the presence of a large number of exposed Nb active sites on Nb₂C supports, the Pd species were anchored on Nb₂C via spontaneous galvanic replacement reaction, thus constructing the system of USMSI. Comprehensive characterizations, DFT calculations and molecular dynamic (MD) simulations evidence that Nb₂C possess stronger interaction with Pd atoms than the Pd-Pd binding, so Pd species were anchored on Nb₂C as metallenes. Nb₂C compel the Pd atoms to diffuse on the support to form 2D metal metallenes instead of 3D metal nanoparticles. However, the functional groups (such as Cl, Br, O) on the surface of Nb₂C attenuated the interaction between metal and support to be smaller than the Pd-Pd interaction in nanoparticle, thus forming a 3D structure of Pd nanoparticles. To distinguish these two categories, USMSI was adopted in this work to describe the interaction between Pd and Nb₂C.

In our previous research¹⁴, the relationship between metal-support interaction (MSI) and contact angle has been built for supported Pt nanoparticle (NP). As shown in **Fig. R4**, for the system of weak MSI (such as Pt NP supported on graphene), the contact angle is larger than 90°. When the MSI enhances and becomes a strong MSI (such as Pt NP supported on Ti₃C₂O₂), the contact angle is less than 90°. As the MSI get stronger and becomes an ultra-strong MSI (USMSI) (such as Pt NP supported on Ti₃C₂), in which Ti₃C₂ possess stronger interaction with Pt atoms than the Pt-Pt binding, the Pt nanoparticle exhibits a planar structure. This mechanism of transforming 3D nanoparticle into 2D layered structures have also been verified by the fabrication of Pd metallenes on Nb₂C in this work.

Fig. R4. Structures of Pt nanoparticles supported on (a) graphene, (b) Ti₃C₂O₂, and (c) Ti₃C₂, respectively.

As the comments by the reviewer, the manuscript is revised as follow:

◆ In the manuscript:

The SMSI can effectively reduce the high surface energy of the metal nanoparticles for one thing. Meanwhile, once the selected supports possess stronger interaction with metal atoms than the metal-metal binding, ultra-strong metal-support interaction (USMSI) is constructed^{25,26}. The supports would compel the metal atoms to spread out on the support to form two-dimensional (2D) metal metallenes instead of three-dimensional (3D) metal nanoparticles, breaking the tradeoff between the catalytic stability and activity of traditional SMSI.

Reference

14. Qiu, C., *et al.* Multiscale Simulation of Morphology Evolution of Supported Pt Nanoparticles via Interfacial Control. *Langmuir* **35**, 6393-6402 (2019). <https://doi.org/10.1021/acs.langmuir.9b00129>.

Comment 7:

The amount of Br according to the XPS is much lower than Cl in the treated supports. Please explain. Any quantification?

Reply: Thanks a lot for your careful guidance. The introduction of Cl and Br is to stir Nb₂C with 20 mL HCl (concentration: 36%) and 20 mL HBr (concentration: 48%) for 2 h, respectively. To probe the contents of Cl and Br, ICP-MS analysis was conducted with Aglient 7800 spectrometer. The supplementary ICP-MS results show that the contents of Br and Cl were 1.1 wt. % and 2.52 wt. %, respectively.

As the comments by the reviewer, the manuscript is revised as follow:

◆In the manuscript:

Methods: The synthetic procedures of Cl-Nb₂C, Br-Nb₂C, O-Nb₂C, Pd/Cl-Nb₂C, Pd/Br-Nb₂C, Pd/O-Nb₂C are presented in Supplementary Information. For Pd/Cl-Nb₂C and Pd/Br-Nb₂C, the contents of Cl and Br were measured to be 2.52 wt. % and 1.1 wt. %, respectively according to the results of ICP-MS.

Comment 8:

The text states that an increase in Pd loading produces further agglomeration, thus losses of Pd metallenes and catalytic activity. However, Fig S10 clearly shows that the yield and selectivity are exactly the same for the 0.3 wt% and the 5 wt% materials. The TEM photographs do not show, to me, higher agglomeration but, perhaps, Pd particle superpositions by the higher population of Pd metallenes. Could be that true? That will be good news, since lower amounts of solid will catalyze the reaction similarly. If not, and Pd nanoparticles are really formed, XRD measurements should show them.

Reply: Thanks a lot for your enlightening comment. **Supplementary Fig. 20** showed the catalytic performance of Pd/Nb₂C with different Pd loadings. To guarantee the same conversion, the reaction time for different catalysts was different. The specific reaction conditions were listed in the following **Table R2**. It was obvious that the increase in Pd loading allowed the conversion of phenylacetylene to be completed in a shorter time. The detail reaction conditions were supplemented in the revised figure.

Based on the reviewers' valuable suggestions, XRD, AFM and MD simulations were supplemented to determine the accurate structure of 5% Pd/Nb₂C. According to TEM images (**Supplementary Fig. 13d**), 5% Pd/Nb₂C displayed the petal shape. The XRD pattern of 5% Pd/Nb₂C showed that no obvious Pd diffraction peaks were detected (**Supplementary Fig. 21**). As proposed by the reviewer, it was reasonable to speculate that the Pd petals were formed by stacking several layers of Pd metallenes rather than Pd nanoparticles. To illustrate this conjecture, AFM analysis was used to measure the thickness of petal Pd on Nb₂C. AFM images confirmed the thickness of 1.67~2.32 nm for the petal Pd species, several times the thickness of Pd metallenes in the 0.5% Pd/Nb₂C (**Supplementary Fig. 22**). Besides, the morphological changes of more Pd atoms (Pd₉₂₃) on the Nb₂C surrounded by O group were further studied by MD simulations. It clearly showed that excess Pd atoms tended to accumulate on the top of Pd metallenes under USMSI between Nb₂C and Pd (**Supplementary Fig. 2**).

Table. R2 | Performance of Pd/Nb₂C with different Pd loadings.

Entry	Catalyst	Catalyst /Substrate (mol %)	Pressure (MPa)	Time (min)	Conv. (%)	Sel. (%)
-------	----------	-----------------------------	----------------	------------	-----------	----------

1	0.3%Pd/Nb ₂ C	0.0138	0.1	55	96	92
2	0.5%Pd/Nb ₂ C	0.023	0.1	40	99	96
3	2%Pd/Nb ₂ C	0.092	0.1	12	96	95
4	5%Pd/Nb ₂ C	0.23	0.1	7	99	92

Reaction conditions: 2 mmol of phenylacetylene, 10 mg catalysts, 5 mL of ethanol, 298 K, 0.1 MPa of H₂.

As the comments by the reviewer, the manuscript is revised as follow:

◆ In the Supplementary Information

Revised Original Supplementary Fig. 17:

Supplementary Fig. 20 | Performance of Pd/Nb₂C with different Pd loadings. Reaction conditions: 2 mmol of phenylacetylene, 10 mg of catalysts, 5 mL of ethanol, 298 K, 0.1 MPa of H₂. The catalytic behaviors of Pd/Nb₂C with different Pd loadings from 0.3 wt. % to 5 wt. % were evaluated by the hydrogenation of phenylacetylene, and these reactions were completed within 55 min, 40 min, 12 min, and 7 min, respectively.

Added Figures and Movies in Supplementary Information

◆ In the Supplementary Information

Supplementary Fig. 2 | Snapshots and energies of Pd₉₂₃ nanoparticles supported on substrate of Nb₂C surrounded by O group for different sites, 923 atoms refer to all Pd atoms on Nb₂C including green, yellow and orange Pd atoms. (a) green Pd clusters anchor on the Nb₂C surrounded by O group. (b) green Pd atoms accumulate as layer on the Pd metal surfaces.

movie1.mp4, movie2.mp4, movie3.mp4

Supplementary Movie 1 | The migration process of Pd nanoparticles when the initial position of the particle is at the boundary of the O functional group.

Supplementary Fig. 21 | XRD pattern of the 5% Pd/Nb₂C.

Supplementary Fig. 22 | AFM images of 5% Pd/Nb₂C. (a, c, e) 2D AFM images of 5% Pd/Nb₂C. (b, d, f) Height profiles along the marked line in (a), (c) and (e).

◆ In the Manuscript:

The different coordination environment of Pd will lead to significant differences in catalytic

performances. The semihydrogenation of phenylacetylene was chosen as a probe reaction to evaluate the catalytic performance for a series of Pd/MXenes. Significantly, Pd/Nb₂C steered the reaction selectively towards styrene even at a conversion of 99% (96% selectivity) at 298 K with 0.1 MPa H₂ (**Fig. 5a**). In contrast to Pd/Nb₂C, the over-hydrogenation of styrene to ethylbenzene was more inclined to take place on Pd supported on Nb₂C modified with functional groups. Selectivities towards styrene on the Pd/Cl-Nb₂C, Pd/Br-Nb₂C and Pd/O-Nb₂C were 90%, 88%, 82%, respectively. The product distribution with reaction time kinetic profile of Pd/Nb₂C revealed the good selectivity towards alkene throughout the reaction (**Fig. 5b**). Besides, the catalytic performance of phenylacetylene over Pd/Nb₂C with different loadings were also evaluated. With the increase of the Pd loading, the activity of the Pd/Nb₂C increased. The 5% Pd/Nb₂C can achieve 99% conversion in 7 minutes (**Supplementary Fig. 20**), but the corresponding selectivity has slightly decreased to 92%. The XRD pattern of 5% Pd/Nb₂C unfolded that no obvious characteristic peaks of Pd was detected (**Supplementary Fig. 21**). It was reasonable to speculate that the Pd petals were formed by stacking several layers of Pd metallenes rather than Pd nanoparticles. AFM images and height profiles confirmed the thickness of ~2 nm for petal Pd species (about five atomic layers) (**Supplementary Fig. 22**). Correspondingly, the morphology of more Pd atoms (Pd₉₂₃) on the Nb₂C surrounded by O group were further studied by MD simulations, which indicated that excess Pd atoms tended to accumulate on the top of Pd metallenes under the USMSI between Nb₂C and Pd (**Supplementary Fig. 2**).

Comment 9:

Color codes are missing in Figure 4 and Fig S14. Where are the oxygen atoms located?

Reply: Thank you for your careful inspection. We're sorry that the lack of color codes in Fig. 4 and original Supplementary Fig. 14 has impeded your reading. The color codes of the elements had been supplemented in **Fig. 4**. The distribution of oxygen atoms in Pd/Nb₂C had been unfolded by EDX-mapping of HAADF-STEM (**Supplementary Fig. 11**), DFT calculations (**Supplementary Fig. 12**) and MD simulations (**Supplementary Fig. 2**). All these results confirmed that the oxygen atoms were tended to distribute at the interface of Pd metallenes and Nb₂C. In order to facilitate your review, the supplementary information about the position of oxygen element was listed in the **Fig. R5**.

Fig. R5. The located position of O atoms in Pd/Nb₂C. (a) DFT model of O atom at the interface of Pd and Nb₂C. (b) MD simulation of Pd metallenes on the O-Nb₂C_{partial}. (c) HAADF-STEM image of Pd/Nb₂C. Elemental mapping of (d) Pd, (e) O and (f) composite mapping of Pd vs O.

Comment 10:

Figure 5: Figures e, g and I are too small to see anything. Please at least amplify them in the SI. Fig. S18: “i-proH” is not correct, please use i-PrOH or simply isopropanol. Substrate scope: “enol” is not a molecule but a functional group. It would be interesting here to test other hydrogenating functional groups besides Cl to test the selectivity of the Pd metallene catalyst, i.e. aldehyde, ketone or nitro group. Minor things: “Two layers” is “two layers” (several times across the SM), check some spacing in references.

Reply: Thank you very much for your careful comments. We are sorry for some formatting errors that have made your reading difficult. Figure e in Figure 5 were amplified in **Supplementary Fig. 24**. The i-proH was revised to i-PrOH. ‘enol’ was displaced with terminal alkenes with a hydroxyl group. In addition, 16 kinds of substrates were supplemented to evaluate the general scope of Pd/Nb₂C, including halogen-substituted alkyne, aldehyde-substituted alkyne, ketone-substituted alkyne, nitro group-substituted alkyne and so on, which were updated in **Table 1**. More detailed experimental data were listed in **Supplementary Table 9**. As for “Two layers” is “two layers”, original supplementary figures including “Two layers” had been revised and DFT models had been simplified. Some spacing mistakes were revised in references. Thank you.

As the comments by the reviewer, the manuscript is revised as follow:

- ◆ In the Supplementary Information

Supplementary Fig. 24 | NMR data for deuterated phenylacetylene catalyzed by Pd/Nb₂C. NMR data for the products of deuterated phenylacetylene carried out by using Pd/Nb₂C in *i*-PrOH as the solvent, and D₂O as deuterium source. These results also confirmed the existence of hydrogen proton in the reaction mechanism for Pd/Nb₂C. (a) Chemical shift of 1-1.5 ppm in Fig. 5e was amplified. (b) Deuterization reaction diagram of phenylacetylene and deuterization verification. (Red) Phenylacetylene, the peaks around 3.47 ppm were corresponding to H atoms on carbon-carbon triple bond. (Blue) Deuterated phenylacetylene, the peaks around 3.46 ppm decreased in the intensity compared to Red.

Referee 2

Comments:

The submitted manuscript by deals with the use of experimental and theoretical approaches to deliver Pd metallenes on Nb₂C with a focus on boosting the semi-hydrogenation of alkynes (six-membered rings). On the brightest side, most of the experimental measures in the lab seem sound enough to exploit novel catalysts in the hydrogenation of alkyne. However, this referee feels that the computational strategy must be further refined is meaningful conclusions are sought. There are several major corrections to be addressed before recommending that work for publication in Nature Communications.

Comment 1:

I guess that Figures 20 - 23 illustrates optimizations for phenylacetylene, am I right? Figures 24 - 28 are rather than confusing and do not help to compare structures vs. relative energy. However, a major issue appears in the computed profiles. The final release of CH₂CHR to the CH₂CHR (gas) counterpart is associated to a barrier of ca. 2 eV (ca. 190 kJ/mol). That barrier is not specifically discussed in Figure 29. In addition, all other computed barriers are in the range of 0.80 - 1.15 eV (77 - 111 kJ/mol). That numeric values significantly differ from the measured activation energy. As stated by the authors in p. 11 'Reaction rates at different temperatures were measured and the activation energies (E_a) of Pd/Nb₂C were fitted to be 39.5 kJ/mol (Fig. 5c), suggesting the superiority of Pd/Nb₂C among Pd-based catalyst reported previously under similar reaction conditions.' Chemical model and level of theory must be cautiously revised to provide more accurate activation energies.

Q1.1: *I guess that Figures 20-23 illustrates optimizations for phenylacetylene, am I right? Figures 24-28 are rather than confusing and do not help to compare structures vs. relative energy.*

Reply for Q1.1: Thank you for your good proposal. The original Figures S20-23 (revised Figures S26-29) illustrate optimized geometries of intermediates and transition states on Pd (111) and Pd/Nb₂C. Figures S24-28 are free energy profiles on Pd (111) and Pd/Nb₂C according to original data from Table S5 and Table S6. We have removed the useless Figures that do not help to compare structures vs. relative energy.

Q1.2: *“However, a major issue appears in the computed profiles. The final release of CH₂CHR to the CH₂CHR (gas) counterpart is associated to a barrier of ca. 2 eV (ca. 190 kJ/mol). That barrier is not*

specifically discussed in Figure 29. In addition, all other computed barriers are in the range of 0.80 - 1.15 eV (77 -111 kJ/mol). As stated by the authors in p. 11 'Reaction rates at different temperatures were measured and the activation energies (E_a) of Pd/Nb₂C were fitted to be 39.5 kJ/mol (Fig. 5c), suggesting the superiority of Pd/Nb₂C among Pd-based catalyst reported previously under similar reaction conditions.' Chemical model and level of theory must be cautiously revised to provide more accurate activation energies."

Reply for Q1.2: Thank the referee for the suggestion. The suggestion has been taken. Because of the strong interaction between molecule and surface, strong chemisorption of CH₂CHR is observed at low coverage condition in the system of Pd (111) ($E_{ad}=-2.77$ eV) and Pd/Nb₂C ($E_{ad}=-2.11$ eV). To be closer to the realistic conditions, we developed our coverage-dependent model in previous work¹⁵⁻¹⁷, and the most challenging factor coverage effects are explicitly taken into account¹⁸⁻²². With the coverage effect, the better diffusion ability of CH₂CHR ($E_{ad} \approx 0$ eV) on Pd/Nb₂C was observed compared to that on Pd (111) with the value of -0.55 eV. In addition, all other computed major pathway barriers are more reasonable in the range of 0.14-0.88 eV (13.4-84.5 kJ/mol). We have added more discussions in the revised manuscript.

As the comments by the reviewer, the manuscript is revised as follow:

◆ In the manuscript

As the accurate structure of Pd metallenes provides an ideal platform for the study of the structure-property relationship, DFT calculations were performed to further get insights into the molecular-level mechanisms of phenylacetylene hydrogenation over Pd/Nb₂C. The Pd (111) was selected as the benchmark catalyst compared with Pd/Nb₂C. First, the adsorption energies of phenylacetylene (CHCR), styrene (CH₂CHR) and ethylbenzene (CH₃CH₂R) on Pd (111) were much stronger than that on Pd/Nb₂C (Supplementary Fig. 26a). A closer study of the charge distribution of these molecules showed that the value of electrons was nearly the same on Pd (111) and Pd/Nb₂C, respectively, indicating electrons may not be the major factor affecting the adsorption energy of molecules. At the geometric configuration level, the longer dispersion distance (0.302 nm) between two adjacent Pd atoms led to the difference in the effective Pd atoms, substantially affecting the adsorption energy of intermediates. Herein, the effective Pd atoms represent the number of Pd atoms directly bonded with the molecules. From Pd (111) to Pd/Nb₂C, the effective Pd atoms in contact with CH₂CHR decreased from 6 to 5. The larger effective Pd atoms there are, the stronger the molecular adsorption. This phenomenon was also consistent with the weaker chemisorption energy of CHCR

and $\text{CH}_3\text{CH}_2\text{R}$ on Pd/Nb₂C compared to on pure Pd (111) (**Supplementary Fig. 26b**).

Because of the strong interaction between molecule and surface, the most challenging factor, coverage effects, should be taken into account. Based on our developed coverage-dependent model in previous work⁶⁴⁻⁶⁶, the first detailed investigation of the reaction mechanisms for the hydrogenation of CHCR using the DFT-D3 functional and state-of-the-art microkinetic modeling^{64,67,68} was explicitly carried out. All possible reaction channels on Pd (111) and Pd/Nb₂C (**Supplementary Figs. 27-39**) were investigated. The complete elementary steps are displayed in **Supplementary Tables 4-8**. With the coverage effect, the better diffusion ability of CH₂CHR ($E_{\text{ad}} \approx 0$ eV) on Pd/Nb₂C was observed compared to that on Pd (111) with the value of -0.55 eV. The high selectivity of CH₂CHR on Pd/Nb₂C can be explained by higher hydrogenation barrier of CH₂CHR+H→CH₂CH₂R ($E_{\text{a}} = 0.58$ eV) and CH₂CH₂R+H→CH₃CH₂R ($E_{\text{a}} = 0.29$ eV), which indicated that CH₂CHR is likely diffusion rather than further hydrogenation (**Fig. 5g**). Both models (coverage-dependent model and non-coverage model) indicated that the higher reaction rate of CH₂CHR formation was observed in the system of Pd/Nb₂C (**Fig. 5h and Supplementary Fig. 39**). Higher reaction rate of alkynes hydrogenation on Pd/Nb₂C was also confirmed by simulating the hydrogenation of 2-methyl-3-butyn-2-ol (MBY) on Pd (111) and Pd/Nb₂C, respectively (**Supplementary Figs. 40-44**). These findings emphasize the critical role of tripodal Pd metallenes and explain why Pd/Nb₂C was far better than Pd (111) in boosting the catalytic performance, possibly profiting from the dilutive effect of Pd atoms in metallenes that could accelerate the diffusion of CH₂CHR.

Revised Fig. 5:

Fig. 5 Catalytic performance of Pd/Nb₂C MXenes catalyst. (a) Catalytic performance of phenylacetylene on the different Pd/Nb₂C MXenes. Reaction conditions: 5 mL ethanol, 10 mg catalysts, 2 mmol phenylacetylene, T = 298 K, H₂ pressure = 0.1 MPa. (b) Phenylacetylene hydrogenation reaction plots of Pd/Nb₂C. (c) Kinetic curves of Pd/Nb₂C at different temperatures and the activation energy. (d) Primary isotope effect observed for Pd/Nb₂C in phenylacetylene hydrogenation. (e) NMR data for the products of styrene hydrogenation carried out by different Pd catalysts in CD₃OD. (f) Catalytic stability for Pd/Nb₂C. (g) The DFT calculated coverage-dependent free energy diagram for the hydrogenation of phenylacetylene (CHCR) on Pd (111) and Pd/Nb₂C. (h) Simulated major pathways for the hydrogenation of CHCR. Reaction rates are also given for the associated molecular transformations. Reaction rates in red represent on Pd/Nb₂C. Reaction rates in blue mean on Pd (111). The values represent the reaction rates for each elementary step and are given in units of s⁻¹.

Added Reference.

- the Activity and Selectivity for Fischer–Tropsch Synthesis on Co(0001): Microkinetic Modeling with Coverage Effects. *ACS Catal.* **9**, 5957-5973 (2019). <https://doi.org/10.1021/acscatal.9b01150>
16. Yao, Z., Zhao, J., Bunting, R. J., Zhao, C., Hu, P., Wang, J. Quantitative Insights into the Reaction Mechanism for the Direct Synthesis of H₂O₂ over Transition Metals: Coverage-Dependent Microkinetic Modeling. *ACS Catal.* **11**, 1202-1221 (2021). <https://doi.org/10.1021/acscatal.0c04125>
 17. Guo, C., Mao, Y., Yao, Z., Chen, J., Hu, P. Examination of the key issues in microkinetics: CO oxidation on Rh(1 1 1). *J. Catal.* **379**, 52-59 (2019). <https://doi.org/10.1016/j.jcat.2019.09.012>
 18. Lausche, A. C., *et al.* On the effect of coverage-dependent adsorbate–adsorbate interactions for CO methanation on transition metal surfaces. *J. Catal.* **307**, 275-282 (2013). <https://doi.org/10.1016/j.jcat.2013.08.002>
 19. Grabow, L. C., Hvolbæk, B., Nørskov, J. K. Understanding Trends in Catalytic Activity: The Effect of Adsorbate–Adsorbate Interactions for CO Oxidation Over Transition Metals. *Topics in Catalysis* **53**, 298-310 (2010). <https://doi.org/10.1016/j.jcat.2013.08.002>
 20. Yang, N., *et al.* Intrinsic Selectivity and Structure Sensitivity of Rhodium Catalysts for C₂+ Oxygenate Production. *J. Am. Chem. Soc.* **138**, 3705-3714 (2016). <https://doi.org/10.1021/jacs.5b12087>
 21. Pilot, I. a. W., Van Santen, R. A., Hensen, E. J. M. The Optimally Performing Fischer-Tropsch Catalyst. *Angew. Chem.* **126**, 12960-12964 (2014). <https://doi.org/10.1002/ange.201406521>
 22. Medford, A. J., *et al.* CatMAP: A Software Package for Descriptor-Based Microkinetic Mapping of Catalytic Trends. *Catal. Lett.* **145**, 794-807 (2015). <https://doi.org/10.1007/s10562-015-1495-6>

Comment 2:

Related to the previous comment, one would expect a more complete assessment of the alkyne semi-hydrogenation performance of the catalysts. In other words, thermodynamic corrections are missing in the present version.

Reply: Thank the referee for the suggestion. We have done the thermodynamic corrections. More details had been added in the Supplementary Information. Vibrational frequency analyses were performed to confirm the integrity of initial states, transition states and final states. The zero-point energy (ZPE) correction was calculated as follows:

$$\text{ZPE} = \sum_i \frac{h\nu_i}{2} \quad (1)$$

where h is Planck's constant and the standard molar vibrational internal energy contribution is calculated as follows:

$$U_{\text{vib}}^0 = RT \sum_i \frac{h\nu_i/K_B}{e^{h\nu_i/K_B T} - 1} \quad (2)$$

where K_B is the Boltzmann constant and R is the gas constant. The standard molar vibrational entropy is given by:

$$S_{\text{vib}}^0 = R \sum_i \left[\frac{\frac{h\nu_i}{K_B T}}{e^{\frac{h\nu_i}{K_B T}} - 1} - \ln(1 - e^{-h\nu_i/K_B T}) \right] \quad (3)$$

Overall, adding the free energy corrections together, the standard molar Gibbs free energy change for the elementary reaction for the hydrogenation reaction can be written as follow:

$$\Delta G^0 = \Delta E + \Delta ZPE + \gamma RT \left(1 + \ln \frac{P}{P_0} \right) + \Delta U^0 - T \Delta S^0 \quad (4)$$

where ΔE represents the difference of the total energies from the VASP calculation. If it is gaseous molecule, then 1 is chosen as the value of γ and 0 is selected for γ for surface reactant and P is the partial pressure.

The free-energy corrections for molecules in the gas phase were calculated using the Gaussian 09 package²³, and the basis set was B3LYP/6-311g^{24,25}.

Added Reference.

23. Gaussview, V., Dennington, Roy; Keith, Todd; Millam, John. Semichem Inc., Shawnee Mission, Ks, 2009.
24. Lee, C., Yang, W., Parr, R. G. Development of the Colle-Salvetti correlation-energy formula into a functional of the electron density. *Phys. Rev. B* **37**, 785-789 (1988). <https://doi.org/10.1103/PhysRevB.37.785>
25. Becke, A. D. Density - functional thermochemistry. III. The role of exact exchange. *J. Chem. Phys.* **98**, 5648-5652 (1993). <https://doi.org/10.1063/1.464913>

Comment 3:

TOF values and the yields of products supported on the selected seven Pd catalysts demonstrated the impact of decoration. The quality and novelty of the paper might be increased by performing additional calculations, e.g., by including one additional substrate (i.e. Pd/V₂O₅, Pd/Al₂O₃) and/or by simulating one additional alkyne.

Reply: Thank the referee for the suggestion. The suggestion has been taken, and the quality and novelty of the paper are increased by performing calculations for one additional alkyne, 2-methyl-3-butyn-2-ol (MBY), on Pd (111) and Pd/Nb₂C, respectively. The calculated optimized intermediates and transition states were shown in **Supplementary Figs. 40-43**. We investigated all possible reaction channels on Pd (111) and Pd/Nb₂C. The higher reaction rate of CH₂CHR formation was observed in the system of Pd/Nb₂C (**Supplementary Fig. 44**). CHCR probably hydrogenates into CRCH₂ first and subsequently forms CH₂CHR. Next, the CH₂CHR is possible diffusion in the gas phase rather than further hydrogenation to paraffin. This conclusion is still valid being consistent with our findings.

Supplementary Fig. 40 | Top views of all the optimized geometries of intermediates on Pd (111) for the hydrogenation of 2-methyl-3-butyn-2-ol (MBY). (a) $\text{CH}_2\text{CH}_2\text{R}$; (b) CH_2CHR ; (c) $\text{CH}_3\text{CH}_2\text{R}$; (d) CHCH_2R ; (e) CHCHR ; (f) CHCR ; (g) CHRCH_3 ; (h) CRCH_2 ; (i) CRCH_3 .

Supplementary Fig. 41 | Top views of calculated transition state geometries on Pd (111) for the hydrogenation of MBY. (a) $\text{CH}_2\text{CH}_2\text{R}+\text{H} \rightleftharpoons \text{CH}_3\text{CH}_2\text{R}$; (b) $\text{CH}_2\text{CHR}+\text{H} \rightleftharpoons \text{CH}_2\text{CH}_2\text{R}$; (c) $\text{CH}_2\text{CHR}+\text{H} \rightleftharpoons \text{CHRCH}_3$; (d) $\text{CHCH}_2\text{R}+\text{H} \rightleftharpoons \text{CH}_2\text{CH}_2\text{R}$; (e) $\text{CHCHR}+\text{H} \rightleftharpoons \text{CH}_2\text{CHR}$; (f) $\text{CHCHR}+\text{H} \rightleftharpoons \text{CHCH}_2\text{R}$; (g) $\text{CHCR}+\text{H} \rightleftharpoons \text{CHCHR}$; (h) $\text{CHCR}+\text{H} \rightleftharpoons \text{CRCH}_2$; (i) $\text{CHRCH}_3+\text{H} \rightleftharpoons \text{CH}_3\text{CH}_2\text{R}$; (j) $\text{CRCH}_2+\text{H} \rightleftharpoons \text{CH}_2\text{CHR}$; (k) $\text{CRCH}_2+\text{H} \rightleftharpoons \text{CRCH}_3$; (l) $\text{CRCH}_3+\text{H} \rightleftharpoons \text{CHRCH}_3$.

Supplementary Fig. 42 | Top views of all the optimized geometries of intermediates on Pd/Nb₂C for the hydrogenation of MBY. (a) CH₂CH₂R; (b) CH₂CHR; (c) CH₃CH₂R; (d) CHCH₂R; (e) CHCHR; (f) CHCR; (g) CHRCH₃; (h) CRCH₂; (i) CRCH₃.

Supplementary Fig. 43 | Top views of calculated transition state geometries on Pd/Nb₂C for the hydrogenation of MBY. (a) CH₂CH₂R+H ⇌ CH₃CH₂R; (b) CH₂CHR+H ⇌ CH₂CH₂R; (c) CH₂CHR+H ⇌ CHRCH₃; (d) CHCH₂R+H ⇌ CH₂CH₂R; (e) CHCHR+H ⇌ CH₂CHR; (f) CHCHR+H ⇌ CHCH₂R; (g) CHCR+H ⇌ CHCHR; (h) CHCR+H ⇌ CRCH₂; (i) CHRCH₃+H ⇌ CH₃CH₂R; (j) CRCH₂+H ⇌ CH₂CHR; (k) CRCH₂+H ⇌ CRCH₃; (l) CRCH₃+H ⇌ CHRCH₃.

Supplementary Fig. 44 | (a) Reaction pathways for the hydrogenation of MBY on Pd (111). (b) Reaction pathways for the hydrogenation of MBY on Pd/Nb₂C. Red arrows represent the important pathways according to the results of the microkinetic model. The values represent the reaction rates for each elementary step and the unit is s⁻¹. (T=298.15 K)

Comment 4:

Authors claimed that the reported results ‘explicitly demonstrated that Pd/Nb₂C is a universal catalyst and displays superior selectivity in the hydrogenation of alkyne.’ Unfortunately, the selected library of alkynes is too narrow to confirm such ‘universality’. A wider panel of alkynes with a larger heterogenicity. This is a must to confirm that the methodology may also be applied to other types of alkynes, which is especially critical for problematic functionalities including ketones, aldehydes, heterocycles, nitrogen and sulfur-containing alkynes, to cite a few.

Reply: Thanks a lot for your critical question. To confirm the universality of Pd/Nb₂C, in addition to the original 12 substrates, a total of 16 kinds of alkynes including aldehyde substituted alkyne, ketone substituted alkyne, nitro group substituted alkyne, heterocyclic alkynes, sulfur-containing alkynes and so on were supplemented in the **Table 1**. The detailed experimental parameters of the hydrogenation reactions and GC-MS characterization for products were listed in the **Supplementary Table 9** and **Supplementary Figs. 45-60**. The results highlight again Pd//Nb₂C was a universal catalyst and displayed superior selectivity in the hydrogenation of various alkynes.

As the comments by the reviewer, the manuscript is revised as follow:

◆ In the manuscript:

Having proved that Pd/Nb₂C was an efficient catalyst, subsequently, to explore the universality of catalysts, we tested the general scope of the catalyst for the hydrogenation of various structurally different substituted alkynes (**Table 1, Supplementary Figs. 45-60 and Supplementary Table 9**). Gratifyingly, the Pd/Nb₂C was found to exhibit consistent selectivity for substituted alkynes. For terminal alkynes with alkyl, hydroxyl, ether, and amino substituents, Pd/Nb₂C delivered more than 90% selectivity towards terminal alkenes (**Table 1, entries 1-11**). Notably, terminal alkynes substituted with halogen groups (-F, -Cl, -Br) were readily hydrogenated to the desired alkenes and no dehalogenation products were detected (**Table 1, entries 12-15**). Pd/Nb₂C also exhibited high alkenes selectivity for the alkynes bearing biphenyl backbones, heterocyclic frameworks, and carbonyl functional groups (**Table 1, entries 16-23**). Impressively, reducible functional groups such as -NO₂, -CHO groups, remained completely unaffected during the hydrogenation process (**Table 1, entries 24, 25**). In addition to terminal alkynes, Pd/Nb₂C can accomplish the smooth hydrogenation of internal alkynes to cis-alkenes with a selectivity of up to 96% for cis-2-buten-1-ol and 1-phenyl-1-propyne (**Table 1, entries 26, 27**). Similarly, Pd/Nb₂C displayed impressive chemoselectivity in the transformation of mifepristone to steroidal drug aglepristone (**Table 1, entry 28**), further indicating the potential application of Pd/Nb₂C in fine chemical industry. These results explicitly demonstrated that Pd/Nb₂C is a universal catalyst and displays superior selectivity in the hydrogenation of alkynes.

Original:

Table 1. The performance of Pd/Nb₂C catalyst toward different substrates

1. 	2. 	3. 	4. 94% (95%)	96% (92%)	94% (92%)	99% (91%)
5. 	6. 	7. 	8. 96% (93%)	92% (95%)	99% (92%)	97% (92%)
9. 	10. 	11. 	12. 93% (92%)	96% (91%)	96% (96%)	97% (96%)

The catalytic evaluation was performed with 2 mmol substrate, 5 mL ethanol and 10 mg Pd/Nb₂C with Pd loading of 0.5 wt%.

Revised Table 1:

Table 1. The performance of Pd/Nb₂C catalyst toward different substrates

1.  94 % (95%)	2.  96 % (92%)	3.  95 % (94%)	4.  96 % (94%)	5.  96% (93%)
6.  92% (93%)	7.  99 % (92%)	8.  97 % (92%)	9.  96% (91%)	10.  92% (95%)
11.  99% (93%)	12.  99% (91%)	13.  94% (92%)	14.  98% (93%)	15.  96% (93%)
16.  92% (95%)	17.  93% (92%)	18.  99% (96%)	19.  88% (89%)	20.  99% (88%)
21.  87% (94%)	22.  99% (88%)	23.  99% (94%)	24.  99% (93%)	25.  99% (92%)
	26.  96% (96%)	27.  97% (96%)	28.  97% (90%)	

Reaction conditions: alkyne substrates, Pd/Nb₂C, 5 mL of ethanol. Detailed reaction conditions are listed in the Supplementary Information. Conversions were reported, and the data in parentheses were the alkenes selectivity.

◆ In the Supplementary Information

Added Table in Supplementary Information

Supplementary Table 9. Data and experimental conditions for substrate hydrogenation in Table 1.

Entry	Substrate	Pressure (MPa) /Temperature (K)	Pd /Substrate (mol %)	Time (min)	Con v. (%)	Sel. (%)
1		0.1/298	0.023	12	94	95
2		0.1/298	0.023	30	96	92
3		0.1/298	0.023	55	95	94
4		0.1/298	0.023	55	96	94
5		0.2/298	0.023	20	96	93
6		0.1/303	0.047	50	92	93
7		0.2/298	0.023	11	99	92
8		0.2/298	0.023	45	97	92

9		0.1/303	0.023	20	96	91
10		0.1/298	0.023	25	92	95
11		0.1/298	0.023	60	99	93
12		0.2/298	0.023	20	99	91
13		0.2/298	0.023	25	94	92
14		0.1/298	0.023	80	98	93
15		0.1/298	0.023	100	96	93
16		0.1/303	0.023	60	92	95
17		0.1/303	0.023	26	93	92
18		0.1/303	0.047	10	99	96
19		0.1/303	0.047	40	88	89

20		0.1/303	0.115	30	99	88
21		0.1/313	0.023	180	87	94
22		0.1/313	0.047	40	99	88
23		0.1/298	0.023	35	99	94
24		0.1/313	0.047	180	99	93
25		0.1/298	0.047	180	99	92
26		0.2/298	0.023	59	96	96(cis)
27		0.2/298	0.023	55	97	96(cis)
28		0.2/303	0.200	600	97	90(cis)

These substrates experiments are all used 5 mL ethanol as solvent, Pd/Nb₂C.

◆ In the supplementary information

Added figures in Supplementary Information:

GC-MS characterization of some supplemental substrates

Supplementary Fig. 45 | GC-MS of entries 3

Supplementary Fig. 46 | GC-MS of entries 4

Supplementary Fig. 47 | GC-MS of entries 6

Supplementary Fig. 48 | GC-MS of entries 11

Supplementary Fig. 49 | GC-MS of entries 14

Supplementary Fig. 50 | GC-MS of entries 15

Supplementary Fig. 51 | GC-MS of entries 16

Supplementary Fig. 52 | GC-MS of entries 18

Supplementary Fig. 53 | GC-MS of entries 19

Supplementary Fig. 54 | GC-MS of entries 20

Supplementary Fig. 55 | GC-MS of entries 21

Supplementary Fig. 56 | GC-MS of entries 22

Supplementary Fig. 57 | GC-MS of entries 23

Supplementary Fig. 58 | GC-MS of entries 24

Supplementary Fig. 59 | GC-MS of entries 25

Supplementary Fig. 60 | LC-MS of entries 28

References

1. Duan, H., *et al.* Ultrathin rhodium nanosheets. *Nat. Commun.* **5**, 3093 (2014).
2. Li, X., *et al.* PdFe Single-Atom Alloy Metallene for N₂ Electroreduction. *Angew. Chem. Int. Ed* **134**, e202205923 (2022).
3. Jiang, J., Ding, W., Li, W., Wei, Z. Freestanding Single-Atom-Layer Pd-Based Catalysts: Oriented Splitting of Energy Bands for Unique Stability and Activity. *Chem* **6**, 431-447 (2020).
4. Jia, Z., *et al.* Fully-exposed Pt-Fe cluster for efficient preferential oxidation of CO towards hydrogen purification. *Nat. Commun.* **13**, 6798 (2022).
5. Xiong, Y., Yang, Y., Disalvo, F. J., Abruña, H. D. Pt-Decorated Composition-Tunable Pd-Fe@Pd/C Core-Shell Nanoparticles with Enhanced Electrocatalytic Activity toward the Oxygen Reduction Reaction. *J. Am. Chem. Soc.* **140**, 7248-7255 (2018).
6. Wang, Z.-S., Yang, C.-L., Xu, S.-L., Nan, H., Shen, S.-C., Liang, H.-W. Electronic Modulation of Pd-Based Bimetallic Catalysts with Sulfur-Doped Carbon Support for Phenylacetylene Semihydrogenation. *Inorganic Chemistry* **59**, 5694-5701 (2020).
7. Cordoba, M., Coloma-Pascual, F., Quiroga, M. E., Lederhos, C. R. Olefin Purification and Selective Hydrogenation of Alkynes with Low Loaded Pd Nanoparticle Catalysts. *Industrial & Engineering Chemistry Research* **58**, 17182-17194 (2019).
8. Weerachawanasak, P., Mekasuwandumrong, O., Arai, M., Fujita, S.-I., Praserttham, P., Panpranot, J. Effect of strong metal-support interaction on the catalytic performance of Pd/TiO₂ in the liquid-phase semihydrogenation of phenylacetylene. *J. Catal.* **262**, 199-205 (2009).
9. Tokmic, K., Fout, A. R. Alkyne Semihydrogenation with a Well-Defined Nonclassical Co-H₂ Catalyst: A H₂ Spin on Isomerization and E-Selectivity. *J. Am. Chem. Soc.* **138**, 13700-13705 (2016).
10. Choi, H. C., Shim, M., Bangsaruntip, S., Dai, H. Spontaneous Reduction of Metal Ions on the Sidewalls of Carbon Nanotubes. *J. Am. Chem. Soc.* **124**, 9058-9059 (2002).
11. Bruno, J. E., *et al.* Supported Ni-Au Colloid Precursors for Active, Selective, and Stable Alkyne Partial Hydrogenation Catalysts. *ACS Catal.* **10**, 2565-2580 (2020).
12. Qu, L., Dai, L. Substrate-Enhanced Electroless Deposition of Metal Nanoparticles on Carbon Nanotubes. *J. Am. Chem. Soc.* **127**, 10806-10807 (2005).
13. Corma, A., Garcia, H., Leyva, A. Catalytic activity of palladium supported on single wall carbon nanotubes compared to palladium supported on activated carbon: Study of the Heck and Suzuki couplings, aerobic alcohol oxidation and selective hydrogenation. *J. Mol. Catal. A* **230**, 97-105 (2005).

14. Qiu, C., *et al.* Multiscale Simulation of Morphology Evolution of Supported Pt Nanoparticles via Interfacial Control. *Langmuir* **35**, 6393-6402 (2019).
15. Yao, Z., Guo, C., Mao, Y., Hu, P. Quantitative Determination of C–C Coupling Mechanisms and Detailed Analyses on the Activity and Selectivity for Fischer–Tropsch Synthesis on Co(0001): Microkinetic Modeling with Coverage Effects. *ACS Catal.* **9**, 5957-5973 (2019).
16. Yao, Z., Zhao, J., Bunting, R. J., Zhao, C., Hu, P., Wang, J. Quantitative Insights into the Reaction Mechanism for the Direct Synthesis of H₂O₂ over Transition Metals: Coverage-Dependent Microkinetic Modeling. *ACS Catal.* **11**, 1202-1221 (2021).
17. Guo, C., Mao, Y., Yao, Z., Chen, J., Hu, P. Examination of the key issues in microkinetics: CO oxidation on Rh(1 1 1). *J. Catal.* **379**, 52-59 (2019).
18. Lausche, A. C., *et al.* On the effect of coverage-dependent adsorbate–adsorbate interactions for CO methanation on transition metal surfaces. *J. Catal.* **307**, 275-282 (2013).
19. Grabow, L. C., Hvolbæk, B., Nørskov, J. K. Understanding Trends in Catalytic Activity: The Effect of Adsorbate–Adsorbate Interactions for CO Oxidation Over Transition Metals. *Topics in Catalysis* **53**, 298-310 (2010).
20. Yang, N., *et al.* Intrinsic Selectivity and Structure Sensitivity of Rhodium Catalysts for C₂+ Oxygenate Production. *J. Am. Chem. Soc.* **138**, 3705-3714 (2016).
21. Pilot, I. a. W., Van Santen, R. A., Hensen, E. J. M. The Optimally Performing Fischer–Tropsch Catalyst. *Angew. Chem. Int. Ed.* **53**, 12746-12750 (2014).
22. Medford, A. J., *et al.* CatMAP: A Software Package for Descriptor-Based Microkinetic Mapping of Catalytic Trends. *Catalysis Letters* **145**, 794-807 (2015).
23. Gaussview, V., Dennington, Roy; Keith, Todd; Millam, John. Semichem Inc., Shawnee Mission, Ks, 2009.
24. Lee, C., Yang, W., Parr, R. G. Development of the Colle-Salvetti correlation-energy formula into a functional of the electron density. *Phys. Rev. B* **37**, 785-789 (1988).
25. Becke, A. D. Density - functional thermochemistry. III. The role of exact exchange. *J. Chem. Phys.* **98**, 5648-5652 (1993).

REVIEWERS' COMMENTS

Reviewer #1 (Remarks to the Author):

The new version of the manuscript covers most of the questions posed by this Reviewer. However, some points must be revised before publication:

- The widen substrate scope is meritorious, however, the lack of characterization of the products beyond GC-MS hampers the credibility of the structures when a second reducible functional group is present in the starting substrate. This is critical for substrates 22-24 and 28, since the hydrogenation of the ketone/aldehyde will give exactly the same product mass than the semi-hydrogenation of the alkyne. Authors should do here NMR or measure the purchased products (as they did for the internal alkenes). For substrate 28, NMR characterization is mandatory, since not only the ketone group but also alkene groups can receive the H₂ molecule, and the final enol group can tautomerize to the aldehyde form.
- The Pd mol% in the catalytic experiments must be indicated in all catalytic procedures, particularly in the general procedure (Methods in the manuscript).
- Supplementary Fig 20 is still unclear for the Reader, since it still gives the perception that the Pd loading on the solid is irrelevant for the catalytic performance. Commenting the reaction times in the caption is not enough, the inclusion of the time factor must be clearer. I suggest to use here turnover frequency (TOF), since this value will clearly differentiate the better catalytic activity of the Pd metallene.
- If the term "ultra-strong metal-support interaction (USMSI)" is firstly proposed in this paper, it must be perfectly defined and emphasized in the text. If not, and it has been previously proposed in similar terms to those here defined, it must be also clearly stated in the main text. Could this effect be quantified by the contact angle? The illustration in Fig. R4 goes in that direction. For a potentially, so relevant, new concept, a separated paper is perhaps recommendable.
- Some explanations about the three supplementary movies provided should be done, at least in the caption.

Reviewer #2 (Remarks to the Author):

Authors have correctly addressed all comments raised by this referee.

1./ Computational details and analysis are now described with a more refined strategy. And even if the selected DFT approach to compute thermodynamics corrections is in the average, it might be accurate enough for such specific goal.

2./ It is also satisfying that the suggestion of expanding the chemical space of testes molecules further confirms the universality of catalysts. Indeed, authors assessed all proposed derivatives, e.g., alkyl, hydroxyl, ether, and amino substituents.

3./ Supplementary Information has been modified/expanded as requested.

In short, authors' efforts are very welcomed as the new data have increased the quality and interest of the original manuscript. In my opinion, this work is suitable for publication in Nature Communications.